# Fractionalization of Coset Non-Invertible Symmetry and Exotic Hall Conductance

Po-Shen Hsin[1,2], Ryohei Kobayashi[3], Carolyn Zhang[4]

[1] *Mani L. Bhaumik Institute for Theoretical Physics, Department of Physics and Astronomy, University of California, Los Angeles, CA 90095, USA*

[2] *Department of Mathematics, King's College London, Strand, London WC2R 2LS, UK.*

[3] *Department of Physics, Condensed Matter Theory Center, and Joint Quantum Institute, University of Maryland, College Park, Maryland 20742, USA*

[4] *Department of Physics, Harvard University, Cambridge, MA02138, USA*

## Abstract

We investigate fractionalization of non-invertible symmetry in (2+1)D topological orders. We focus on coset non-invertible symmetries obtained by gauging non-normal subgroups of invertible 0-form symmetries. These symmetries can arise as global symmetries in quantum spin liquids, given by the quotient of the projective symmetry group by a non-normal subgroup as invariant gauge group. We point out that such coset non-invertible symmetries in topological orders can exhibit symmetry fractionalization: each anyon can carry a "fractional charge" under the coset non-invertible symmetry given by a gauge invariant superposition of fractional quantum numbers. We present various examples using field theories and quantum double lattice models, such as fractional quantum Hall systems with charge conjugation symmetry gauged and finite group gauge theory from gauging a non-normal subgroup. They include symmetry enriched $S_3$ and $O(2)$ gauge theories. We show that such systems have a fractionalized continuous non-invertible coset symmetry and a well-defined electric Hall conductance. The coset symmetry enforces a gapless edge state if the boundary preserves the continuous non-invertible symmetry. We propose a general approach for constructing coset symmetry defects using a "sandwich" construction: non-invertible symmetry defects can generally be constructed from an invertible defect sandwiched by condensation defects. The anomaly free condition for finite coset symmetry is also identified.

June 11, 2024

# 1 Introduction

Symmetry fractionalization describes how zero-form symmetry can act projectively in topologically ordered systems [1, 2]. Given an underlying topological order and an invertible zero-form symmetry, one can enumerate all of the mathematically consistent ways the symmetry can act projectively. In particular, junctions of the symmetry defects can be modified by an Abelian anyon.

Given the enormous body of work on fractionalization of invertible symmetries, it is natural to consider the fractionalization of non-invertible symmetries (see [3–7] for recent reviews of non-invertible symmetries). Examples of non-invertible symmetries are discussed extensively in the literature. They occur in various field theories such as discussed in [8–17] and lattice models such as discussed in [18–31]. In this work we present a first step toward extending symmetry fractionalization to non-invertible symmetries.

As in the case of invertible symmetries, the anomalies of non-invertible symmetry constrain the low energy dynamics [8, 32–36]. Since the anomalies depend on the fractionalization pattern [35–39], it is important to understand how non-invertible symmetry fractionalizes in quantum systems.

We will focus on a class of non-invertible symmetries, given by cosets $G/K$ for group $G$ and a non-normal subgroup $K$ [6, 11, 40–43]. Such coset symmetries arise naturally in spin liquids: when the microscopic model has symmetry $G$, i.e. the "projective symmetry group", a subgroup (the "invariant subgroup") can act trivially at low energy and it is gauged [44–46]. Such models describe $K$ gauge theory of spin liquids, and they have coset global symmetry $G/K$, which is non-invertible for non-normal $K$. Thus, understanding the fractionalization of the coset symmetry is important for such quantum spin liquids.

For instance, in many condensed matter system the $U(1)$ electromagnetism is usually treated as a classical probe field. When the system has charge conjugation symmetry, we can consider the situation where the electromagnetism can be extended to be Alice electromagnetism [47–49] where the $\mathbb{Z}_2$ charge conjugation is gauged. While the fluctuations of the $U(1)$ gauge field can be suppressed by the electric coupling, the fluctuations of the flat $\mathbb{Z}_2$ gauge field are not suppressed, leading naturally to a $(U(1) \rtimes \mathbb{Z}_2)/\mathbb{Z}_2$ electromagnetic coset non-invertible symmetry. This provides further motivation for studying such class of non-invertible symmetry. We will show that such a system with coset symmetry can also have a well-defined electric Hall conductance, that cannot be explained by taking the symmetry to be $U(1)$.

## 1.1 Summary of results

Here we summarize the main results of the paper. We consider the coset non-invertible symmetry expressed as $G/K$, obtained by gauging a discrete subgroup $K$ of the invertible zero-form symmetry $G$. The symmetry $G$ can be either continuous or discrete. We show examples where the coset symmetry is fractionalized in (2+1)D topological phases.

An example of such symmetry fractionalization is found in fractional quantum Hall (FQH) systems with $\mathbb{Z}_2$ charge conjugation symmetry gauged. By FQH system, we simply mean a topological order with a zero-form $U(1)$ symmetry that is fractionalized, leading to anyons carrying fractional $U(1)$ charge. For simplicity, we will consider bosonic FQH systems, but the discussion can be extended to fermionic ones. In FQH states, the $U(1)$ fractionalization is

understood by identifying an Abelian anyon $v$, called the vison, with the vortex of $U(1)$ global symmetry. Fractional $U(1)$ charge of each anyon in the system is determined by its braiding with $v$. In other words, the junction of $U(1)$ symmetry defects is modified by a decoration of an Abelian anyon $v$. When we gauge the charge conjugation of the (bosonic) Abelian FQH state described by $U(1)_{2k}$ Chern-Simons theory, the resulting theory is a non-Abelian topological order described by $O(2)_{2k}$ Chern-Simons theory. The gauged FQH state now has a continuous non-invertible symmetry $(U(1) \rtimes \mathbb{Z}_2)/\mathbb{Z}_2$. This symmetry is referred to as cosine symmetry, since its non-invertible fusion rule is reminiscent of the product to sum formula of $\cos\theta$.

We point out that this cosine symmetry is fractionalized, where the anyons of $O(2)_{2k}$ (for $k = 1$ the anyons are Abelian) carries fractional charge under the continuous non-invertible symmetry. This fractional charge of the anyon is a certain superposition of the opposite fractional charges related by charge conjugation.

Since the conductivity matrix is even under charge conjugation, one can still define the electric Hall conductance in the gauged system, which is now associated with the response under the continuous non-invertible symmetry. While the fractional part of the $U(1)$ electric Hall conductance of the standard (bosonic) FQH state is determined by the spin of the Abelian anyon as $\sigma_H = 2h_v$ mod 2, the vison after gauging the charge conjugation typically behaves as a non-Abelian anyon. Hence, the Hall conductivity is no longer associated with the spin of an Abelian anyon, but rather with the spin of a non-Abelian anyon. Therefore, the value of Hall conductance for continuous non-invertible symmetry cannot generally be computed in the same way as that for an invertible $U(1)$ symmetry. We show that the nonzero Hall conductance enforces the gapless edge state, once the boundary theory is required to preserve the cosine symmetry.

Similar to Abelian FQH states, phases with fractionalization of non-invertible coset symmetry can also be understood from modification of junctions of the zero-form symmetry defects. To describe the junctions of the coset symmetry, we first need to describe the individual defects. We show that the defects can be obtained using a "sandwich" of topological operators. Specifically, the coset symmetry defects are generally described by an invertible symmetry defects sandwiched by the non-invertible topological interfaces. This expression is useful for systematically describing the fractionalization of the coset symmetry using the standard symmetry fractionalization theory of invertible zero-form symmetries [1]. We establish an algebraic formalism of the fractionalization for coset symmetry $G/K$ in (2+1)D bosonic TQFT, in terms of the $G$-crossed braided fusion category interacting with the non-invertible topological interfaces. When $G$ is a finite group, we also find necessary conditions for the coset symmetry $G/K$ being free of 't Hooft anomalies.

We further find that fractionalized coset symmetry can be realized as an exact symmetry of the microscopic lattice model of the non-Abelian topological order. In particular, we find a fractionalized cosine symmetry in the $S_3$ quantum double model [18, 50], which is arguably

the simplest exactly solvable lattice model that hosts non-Abelian topological order. This cosine symmetry of the $S_3$ quantum double model is understood as the resulting symmetry from gauging the $\mathbb{Z}_2$ charge conjugation of $U(1) \rtimes \mathbb{Z}_2$ symmetry in the $\mathbb{Z}_3$ toric code. In the $S_3$ quantum double model, the fractionalized cosine symmetry acts on a non-Abelian anyon to produce the cosine of the non-trivial $U(1)$ fractional charge $q$, reflecting that the non-Abelian anyon carries a superposition of the fractional charge $(q, -q)$.

The work is organized as follows. In section 2 we review the fractionalization of invertible symmetry in terms of Abelian anyons. In section 3, we discuss a class of non-invertible coset symmetry constructed from the outer automorphism, such as $\mathbb{Z}_2$ charge conjugation of $U(1)$. In section 4, we construct quantum double lattice model for $S_3$ enriched by non-invertible coset symmetry. In section 5, we discuss general coset non-invertible symmetry and a bulk TQFT that describes the symmetry, and use the bulk TQFT to derive dynamical consequences from the coset symmetry. In section 6, we apply the discussion to provide construction of new quantum spin liquids enriched with non-invertible coset symmetry, and give examples of new deconfined quantum critical points with non-invertible symmetry. In section 7, we discuss the results and mention several future directions.

## 2 Review of Invertible Symmetry Fractionalizations

Let us first briefly review the fractionalization of invertible symmetry in (2+1)D. The subject is discussed extensively in the literature (e.g. [1, 39, 51–53]), and we will only summarize the properties relevant to our discussion.

### 2.1 Fractionalizations: modifying junctions of symmetry

An invertible $q$-form symmetry is generated by codimension-$(q+1)$ topological defects in Euclidean spacetime[1] that each have an inverse. To fully specify the $q$-form symmetry, we also need to specify the junctions of the domain walls. A codimension $k > q+1$ junction where multiple defects meet can be modified with the topological defects of $(k-1)$-form symmetry. The allowed modification is constrained by the higher-group structure of the symmetries [2]. In this work we will focus on $q = 0$ and $k = 2$. For zero-form symmetry $G$ and 1-form symmetry $\mathcal{A}$, the fractionalizations can be classified by $H^2(G, \mathcal{A})$, which specifies the Abelian anyon at the codimension-two junctions of $G$ defects in (2+1)D [1].[2] If the junction of three $G$ domain walls $g_1, g_2, g_1 g_2$ are modified with additional 1-form symmetry defect $\eta(g_1, g_2) \in \mathcal{A}$, then the line operators that braid with the junction acquire additional projective representation given by the braiding phase with $\eta(g_1, g_2)$. The 1-form symmetry

---

[1]Here, we will use defect and operators interchangeably.

[2]In the presence of 2-group symmetry, some fractionalization classes can be identified [38, 39].

operators inserted at the junction modify the correlation functions of the zero-form symmetry. This can result in additional anomalies depending on the fractionalization (see e.g. [37–39]).

In terms of symmetry operators on the Hilbert space, fractionalization means that the fusion rules of the $q$-form symmetry generators $U_q(\Sigma)$ supported on a codimension-$q$ spatial region with boundary is modified compared with the generators on region without a boundary. The spatial boundary can be modified with another codimension-$(q+1)$ symmetry generator $U_{q+1}(\partial\Sigma)$ for a $(q+1)$-form symmetry. Such boundary modification affects the symmetry action on excitations that have braiding with the $U_{q+1}(\partial\Sigma)$ symmetry.

A method of describing particular fractionalization class is breaking the 1-form symmetry $\mathcal{A}$ with additional microscopic degrees of freedom to screen the 1-form symmetry generator, then the 1-form symmetry generator is no longer topological and we do not have the freedom to modify the junction. Rather, when the 1-form symmetry generator is stuck at the junction the entire configuration is topological. Since there is no 1-form symmetry topological operators that can modify the junction, there is no longer freedom of changing the fractionalization, and the microscopic model without the 1-form symmetry corresponds to a particular fractionalization class. For instance, in gauge theories we can break the center 1-form symmetry by introducing heavy matter fields in the fundamental representation, and we can break the magnetic 1-form symmetry (for instance, consider continuous gauge group in (3+1)D) by introducing dynamical magnetic monopoles, which replace the non-simply connected gauge group such as $U(1)$ with a simply connected gauge group such as $SU(2)$ of the same rank. In TQFT, the fractionalization class is an additional data in TQFT enriched with symmetry.

## 2.2   Example: fractional quantum Hall systems with $U(1)$ symmetry

As an illustrative example, we can describe the fractionalization of $U(1)$ symmetry in FQH systems. For simplicity, we will only consider bosonic FQH systems. These theories can be described as bosonic topological orders with $G = U(1)$ zero-form symmetry.

The $U(1)$ symmetry fractionalization is described by an Abelian anyon $a$ inserted at the junction of three $U(1)$ transformations $\theta_1, \theta_2, [\theta_1 + \theta_2]_{2\pi}$ where $\theta_i \sim \theta_i + 2\pi$ are angles, and $[\theta]_{2\pi}$ is the restriction of $\theta$ to $[0, 2\pi)$. The Abelian anyon can be written as $v^{\eta(\theta_1, \theta_2)}$, where $v$ is an Abelian anyon called a vison, with $\eta(\theta_1, \theta_2) = \frac{[\theta_1]_{2\pi} + [\theta_2]_{2\pi} - [\theta_1 + \theta_2]_{2\pi}}{2\pi}$. [3] This has the following consequences:

- Particles can carry fractional $U(1)$ charge given by the mutual statistics with $v$. The particles that braid with the $v$ with phase $e^{i\phi}$ carry fractional charge given by $\phi/2\pi$

---

[3] In fermionic systems, fractional quantum Hall states obey the spin/charge relation where the $\mathbb{Z}_2$ subgroup of the $U(1)$ electromagnetism is identified with the fermion parity symmetry. This implies that there is an odd number $k$ such that $v^k = f$ is the transparent fermion. For instance, in the fermionic Laughlin state $U(1)_3$, the simplest choice for $v$ has spin 1/6 and $k = 3$.

mod 1: the transformations by $\theta_1, \theta_2$ do not compose into that of $[\theta_1 + \theta_2]_{2\pi}$, but only up to a phase $e^{i\frac{\phi}{2\pi}([\theta_1]_{2\pi} + [\theta_2]_{2\pi} - [\theta_1 + \theta_2]_{2\pi})}$.

- The self-statistics of the Abelian anyon at the junction results in nontrivial correlation function of the $U(1)$ symmetry defects. The fractional quantum Hall conductance $\sigma_H$ for the $U(1)$ symmetry is given by the spin of the Abelian anyon as $\sigma_H = 2h_v \mod 2$ (see e.g. [2]).

# 3 Fractionalization of Non-Invertible Symmetry: An Example

## 3.1 Coset Non-Invertible Symmetry From Outer Automorphism

Let us describe a mechanism for coset non-invertible symmetry. We start with a system with an invertible symmetry $G$, which has automorphism group $\mathrm{Aut}(G)$. $G$ can be either continuous or discrete. We gauge a discrete symmetry $K$ that acts on $G$ by $\rho : K \to \mathrm{Aut}(G)$. If $\rho$ maps elements in $K$ to nontrivial elements of $\mathrm{Aut}(G)$, then the symmetry $G$ becomes non-invertible. To see this, we note that an operator $U_g$ for $g \in G$ is not gauge invariant under discrete gauging if $g$ is permuted by $\rho$. Instead, the well-defined symmetry generator is

$$\tilde{U}_{[g]} = \bigoplus_{k \in K} U_{\rho_k(g)} , \tag{3.1}$$

where $[g]$ is the orbit under the action of $K$. The non-invertible fusion rule of $\tilde{U}$ follows from the invertible fusion rules of $\{U_{\rho_k(g)}\}$ up to condensation defects.[4]

We can describe the non-invertible symmetry as the coset

$$\text{Non-invertible coset symmetry} \quad \tilde{G} = \frac{G \rtimes_\rho K}{K} , \tag{3.2}$$

where the discrete group $K$ acts on $G$ by $\rho$. Since $K$ is not a normal subgroup for nontrivial action $\rho$, the coset is not a group, but rather a non-invertible symmetry. In addition, when $K$ is not normal, the left and right cosets differ. Here, we will always refer to the right coset.

**Example: cosine symmetry from gauging charge conjugation**  Consider $G = U(1)$, and $K = \mathbb{Z}_2$ that acts on $G$ by charge conjugation. Gauging $K$ results in the non-invertible symmetry

$$\tilde{U}_{[\theta]} = U_\theta \oplus U_{-\theta} , \tag{3.3}$$

---

[4]As we will show below, when the fusion of $U_{\rho(g)}$ produces the trivial element in $G$, the fusion outcome is replaced by condensation defect for gauging $K$ symmetry on the domain wall, i.e. condensing the Wilson lines of $K$ on the wall.

where the $U(1)$ transform is $e^{i\theta}$, and $\theta \to -\theta$ under charge conjugation. It satisfies the fusion rule (here we take $\theta \neq \pm\theta'$)

$$\tilde{U}_{[\theta]} \times \tilde{U}_{[\theta']} = \tilde{U}_{[\theta+\theta']} + \tilde{U}_{[\theta-\theta']} \ . \tag{3.4}$$

The above fusion rule is reminiscent of the formula $2\cos\theta\cos\theta' = \cos(\theta+\theta') + \cos(\theta-\theta')$, so this continuous non-invertible symmetry is referred to as cosine symmetry.

The cosine symmetry can be described by the coset $O(2)/\mathbb{Z}_2$, where $\mathbb{Z}_2$ is the charge conjugation. Since the $\mathbb{Z}_2$ is not a normal subgroup, the coset is not a group, but rather a non-invertible symmetry.

**Example: non-invertible SWAP symmetry** Consider two copies of a system with $K$ symmetry, in total the theory has $G = (K \times K) \rtimes \mathbb{Z}_2$ symmetry, where the $\mathbb{Z}_2$ exchanges the two copies and thus swaps the two $K$ groups. Let us then gauge one of the $K$ non-normal subgroup symmetry: this results in the non-invertible coset symmetry

$$\frac{G}{K^{(1)}} = \frac{(K^{(1)} \times K^{(2)}) \rtimes \mathbb{Z}_2}{K^{(1)}} \ , \tag{3.5}$$

where the $K$ symmetry in $j$th layer is written as $K^{(j)}$. In particular, the $\mathbb{Z}_2$ SWAP symmetry becomes non-invertible, with the fusion rule

$$\widetilde{\mathrm{SWAP}} \times \widetilde{\mathrm{SWAP}} = C_{\mathrm{Rep}(K^{(1)})} \left( \sum_{k^{(2)} \in K^{(2)}} U_{k^{(2)}} \right) , \tag{3.6}$$

where $C_{\mathrm{Rep}(K^{(1)})}$ is the condensation defect of the $K^{(1)}$ Wilson lines $\mathrm{Rep}(K^{(1)})$ in the first layer [54], and $U_{k^{(2)}}$ is the generator of global $K^{(2)}$ symmetry in the second layer. An example of non-invertible SWAP symmetry with $K = \mathbb{Z}_2$ is found in [29].

## 3.2 Non-invertible symmetry as sandwich of invertible symmetry

Let us consider the $(d+1)$D theory $\mathcal{T}$ with zero-form symmetry $G$, and gauge the discrete non-normal subgroup $K$. As discussed in section 3.1, the gauged theory $\mathcal{T}/K$ has a non-invertible symmetry $G/K$ with symmetry defects $\tilde{U}_{[g]}$. The symmetry defect $\tilde{U}_{[g]}$ can be expressed as the invertible symmetry defect $U_g$ sandwiched by non-invertible domain walls. To determine the appropriate choice of non-invertible domain walls, we must first determine the "minimal" nonnormal subgroup $K/\tilde{K}$. Specifically, there may be a subgroup $\tilde{K} \subset K$ which is a normal subgroup of $G$. Such a $\tilde{K}$ is obtained by a group of elements $k \in K$ satisfying $gkg^{-1} \in K$ for any $g \in G$, which can be checked to be a normal subgroup of $K, G$. In this case, we first gauge the maximal normal subgroup $\tilde{K}$ of $K$ to obtain an invertible

symmetry $G/\tilde{K}$ and then gauge the remaining part $K/\tilde{K}$.[5] Then the symmetry defect for $G/K$ takes the form

$$\tilde{U}_{[g]} = D_{\mathrm{Rep}(K/\tilde{K})} \times U_g \times \overline{D}_{\mathrm{Rep}(K/\tilde{K})}. \tag{3.7}$$

Here, $U_g$ is a $g \in G/\tilde{K}$ symmetry defect of the theory $\mathcal{T}/\tilde{K}$ before gauging $K/\tilde{K}$. $D_{\mathrm{Rep}(K/\tilde{K})}$ condenses the Wilson line $\mathrm{Rep}(K/\tilde{K})$ of the $K/\tilde{K}$ gauge theory $\mathcal{T}/K = (\mathcal{T}/\tilde{K})/(K/\tilde{K})$, and regarded as a half gauging defect that interpolates $\mathcal{T}/K$ and $\mathcal{T}/\tilde{K}$ [13] (see Figure 1 for the case with trivial $\tilde{K}$). Without loss of generality, we can assume that $K$ is minimal with $\tilde{K} = \{\mathrm{id}\}$, since $G/\tilde{K}$ is a group. In the following we will use condensation defects $D_{\mathrm{Rep}(K)}$. We will shortly see that the above defect $\tilde{U}_{[g]}$ only depends on the orbit $[g]$ of $g \in G$ under the action of $K$. Let us first see a few simple examples for the defect $\tilde{U}_{[g]}$:

- The defect $\tilde{U}_{[g]}$ obviously gives an invertible symmetry defect if and only if $D_{\mathrm{Rep}(K)}$, $\overline{D}_{\mathrm{Rep}(K)}$ are trivial.

- When the invertible defect $U$ in the middle is trivial, the defect $\tilde{U}_{[g]}$ is a condensation defect condensing the electric particles $\mathrm{Rep}(K)$. This corresponds to 1-gauging the algebra object $\mathrm{Rep}(K)$ at a codimension-1 defect [54].

The fusion rules of the half gauging defects are given by (up to normalization)

$$\overline{D}_{\mathrm{Rep}(K)} \times D_{\mathrm{Rep}(K)} = \sum_{k \in K} U_k, \tag{3.8}$$

$$D_{\mathrm{Rep}(K)} \times \overline{D}_{\mathrm{Rep}(K)} = C_{\mathrm{Rep}(K)}, \tag{3.9}$$

where $U_k$ is the $K$ symmetry defect of $\mathcal{T}$ and $C_{\mathrm{Rep}(K)}$ is a condensation defect obtained by 1-gauging $\mathrm{Rep}(K)$. Also

$$U_k \times \overline{D}_{\mathrm{Rep}(K)} = \overline{D}_{\mathrm{Rep}(K)}, \quad D_{\mathrm{Rep}(K)} \times U_k = D_{\mathrm{Rep}(K)}, \tag{3.10}$$

with $k \in K$, and

$$\overline{D}_{\mathrm{Rep}(K)} \times W_\rho = \overline{D}_{\mathrm{Rep}(K)}, \quad W_\rho \times D_{\mathrm{Rep}(K)} = D_{\mathrm{Rep}(K)}, \tag{3.11}$$

where $W_\rho$ is an Wilson line carrying the irreducible representation $\rho \in \mathrm{Rep}(K)$.

From the fusion rule between $U_k$ and $D_{\mathrm{Rep}(K)}$ one can see that the definition of $\tilde{U}_{[g]}$ only depends on the orbit $[g]$,

$$\tilde{U}_{[g]} = D_{\mathrm{Rep}(K)} \times U_g \times \overline{D}_{\mathrm{Rep}(K)} = D_{\mathrm{Rep}(K)} \times U_k U_g U_{k^{-1}} \times \overline{D}_{\mathrm{Rep}(K)} = \tilde{U}_{[kgk^{-1}]}. \tag{3.12}$$

---

[5]We use this convention so that if $K$ is normal, then the defect under the sandwich construction is invertible. If we do not gauge $\tilde{K}$ first, then generically our construction would produce the $G/K$ defect together with a condensation defect, even if $K$ is normal.

The fusion algebra of $\tilde{U}_{[g]}$ follows from the fusion rules above:

$$
\begin{aligned}
\tilde{U}_{[g]} \times \tilde{U}_{[g']} &= D_{\mathrm{Rep}(K)} \times U_g \times \left( \sum_{k \in K} U_k \right) \times U_{g'} \times \overline{D}_{\mathrm{Rep}(K)} \\
&= D_{\mathrm{Rep}(K)} \times \left( \sum_{k \in K} U_{gkg'k^{-1}} U_k \right) \times \overline{D}_{\mathrm{Rep}(K)} \\
&= D_{\mathrm{Rep}(K)} \times \left( \sum_{k \in K} U_{gkg'k^{-1}} \right) \times \overline{D}_{\mathrm{Rep}(K)} \\
&= \sum_{k \in K} \tilde{U}_{[gkg'k^{-1}]}
\end{aligned}
\tag{3.13}
$$

When $[g'] = [g^{-1}]$, one of the fusion channels is $\tilde{U}_{[1]}$ which is the condensation defect

$$
\tilde{U}_{[1]} = C_{\mathrm{Rep}(K)}.
\tag{3.14}
$$

Finally, we can compute the fusion rules of $\tilde{U}_{[g]}$ with the Wilson line of $K$ gauge theory using (3.11):

$$
\tilde{U}_{[g]} \times W_\rho = \tilde{U}_{[g]} = W_\rho \times \tilde{U}_{[g]}.
\tag{3.15}
$$

**Junction of coset symmetry defects**   Two coset symmetry defects $\tilde{U}_{[g]}$ can fuse at the junction into a third defect, which corresponds to one of the fusion outcomes

$$
\tilde{U}_{[g]} \times \tilde{U}_{[g']} = \sum_{k \in K} \tilde{U}_{[gkg'k^{-1}]} \ .
\tag{3.16}
$$

For example, the junction for the fusion channel $\tilde{U}_{[g]} \times \tilde{U}_{[g']} \to \tilde{U}_{[gkg'k^{-1}]}$ can be obtained by the junction of invertible $G$ symmetry defects $g, kg'k^{-1}$ and $gkg'k^{-1}$ sandwiched by the non-invertible defects $D_{\mathrm{Rep}(K)}$, as shown Figure 2 (a).

As mentioned earlier, the sandwich construction for the coset symmetry defect has a redundancy $\tilde{U}_{[g]} = \tilde{U}_{[kgk^{-1}]}$. In other words, the invertible defect $g \in G$ or $kgk^{-1} \in G$ leads to the same coset symmetry defect. This leads to the redundancy in expressing the junction of the coset symmetry defects: the junction of invertible defects $g, g'$ into $gg'$ gives an identical junction of coset symmetry defects as the junction of $kgk^{-1}, kg'k^{-1}$ into $kgg'k^{-1}$ for $k \in K$ (see Figure 2 (b)).

**Example: cosine symmetry**   For example, when $G = O(2) = U(1) \rtimes \mathbb{Z}_2$ and $K = \mathbb{Z}_2 = \{1, c\}$ that acts on $G$ by charge conjugation, the fusion rule of the symmetry defects of $\mathcal{T}/K$ is given by

$$
\begin{aligned}
\tilde{U}_{[\theta]} \times \tilde{U}_{[\theta']} &= D_{\mathrm{Rep}(K)} \times U_\theta \times (1 + U_c) \times U_{\theta'} \times \overline{D}_{\mathrm{Rep}(K)} \\
&= D_{\mathrm{Rep}(K)} \times (U_{\theta+\theta'} + U_{\theta-\theta'} U_c) \times \overline{D}_{\mathrm{Rep}(K)}
\end{aligned}
\tag{3.17}
$$

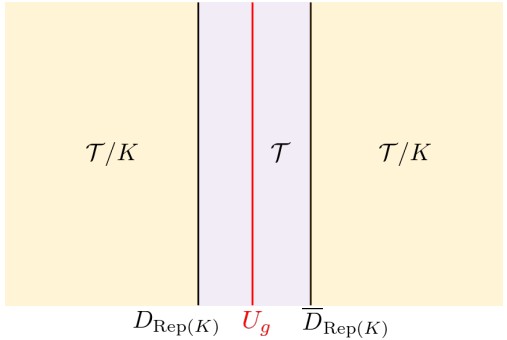

Figure 1: The continuous non-invertible symmetry defect of the gauged theory $\mathcal{T}/K$ is understood as an invertible defect sandwiched by a pair of half gauging defects.

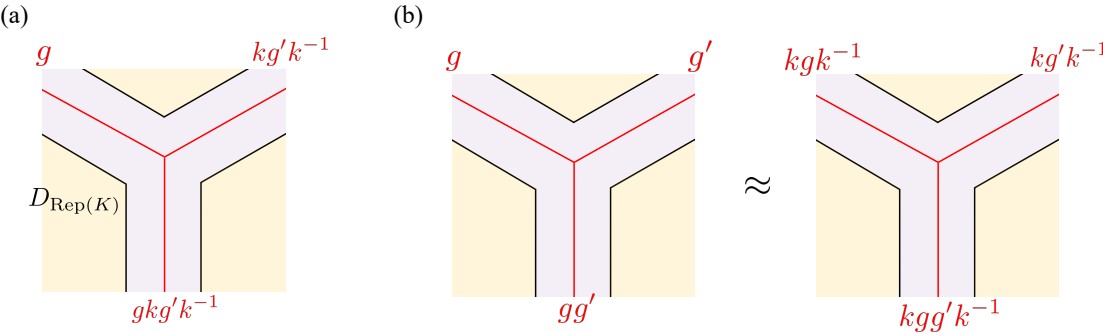

Figure 2: (a): The junction of the cosine symmetry defects that corresponds to the fusion channel $\tilde{U}_{[g]} \times \tilde{U}_{[g']} \to \tilde{U}_{[gkg'k^{-1}]}$. (b): Conjugating the invertible defects by $k \in K$ in the network of coset symmetry defects leads to an another expression of the same defect network in terms of a sandwich.

We then have the fusion rule $U_c \times \overline{D}_{\mathrm{Rep}(K)} = \overline{D}_{\mathrm{Rep}(K)}$, so

$$\tilde{U}_{[\theta]} \times \tilde{U}_{[\theta']} = \tilde{U}_{[\theta+\theta']} + \tilde{U}_{[\theta-\theta']} \tag{3.18}$$

This reproduces the fusion rule of the cosine symmetry (3.4), and is a particular case of (3.13). When $[\theta] = [\theta']$ we get

$$\tilde{U}_{[\theta]} \times \tilde{U}_{[\theta]} = \tilde{U}_{[2\theta]} + C_{\mathrm{Rep}(K)}. \tag{3.19}$$

## 3.3 Exotic quantum Hall conductance for continuous coset symmetry

In the section above, we described how to construct defects of non-invertible 0-form symmetry. We now study the possible decorations of junctions of such defects; these decorations specify fractionalization patterns of non-invertible 0-form symmetry. We begin with an example. Let

us start with an ordinary fractional quantum Hall system with $U(1)$ symmetry in (2+1)D, and then gauge the charge conjugation symmetry. This converts the $U(1)$ symmetry into the non-invertible cosine symmetry, which can be expressed as the coset $O(2)/\mathbb{Z}_2$ for non-normal charge conjugation $\mathbb{Z}_2$. The non-invertible symmetry obeys the fusion rule (3.4). We note that the cosine symmetry does not permute the anyons, just as the ordinary continuous $U(1)$ symmetry.

The current is odd under $\mathbb{Z}_2$ charge conjugation and therefore becomes not gauge invariant. Similarly, the electric and magnetic fields are also odd under $\mathbb{Z}_2$ charge conjugation. Thus the conductivity matrix is gauge invariant after gauging the charge conjugation and well-defined.

Another way to see that the Hall conductance remains well defined is by using the current two point function. This quantity is invariant under charge conjugation, and its contact term gives the Hall conductivity:

$$\langle j_\mu(x, y, t) j_\nu(0) \rangle \supset \sigma_H \epsilon_{\mu\nu\lambda} \partial^\lambda \delta^3(x, y, t) , \tag{3.20}$$

where $\sigma_H$ is the quantum Hall conductance in the unit of $e^2/h$ for electron charge $e$ and Planck constant $h$. The formula follows from taking the functional derivative in the quadratic Chern-Simons response [55]. We note that while the full two-point function requires a Wilson line of the $\mathbb{Z}_2$ gauge field connecting the two points after gauging the charge conjugation symmetry, the contact term is not modified. Thus the Hall response is well-defined for the non-invertible cosine symmetry, and it is given by the response before gauging the charge conjugation symmetry.

### 3.3.1 $O(2)$ Chern-Simons theory

We now show that the theory above has a fractional quantum quantum Hall response $\sigma_H$ for non-invertible continuous symmetry that cannot be obtained from the spin of a vison.

Let us start with $U(1)_{2k}$ theory for integer $k > 1$. We can enrich the theory with $U(1)$ symmetry by identifying the vison with the charge-one Abelian anyon $v$. We can read off the Hall conductance as $\sigma_H = 1/(2k)$, from the spin of $v$.

Now, let us gauge the charge conjugation symmetry in the minimal way. In general, gauging a 0-form symmetry in (2+1)D comes with a choice of an $H^3(K, U(1))$ class. Physically, this corresponds to stacking with a $K$ SPT before gauging. For simplicity, we will only consider gauging without any additional SPT in this example.

The charge conjugation symmetry transforms $a \to a^{-1}$ for a generic Abelian anyon $a$, so the gauged theory becomes a non-Abelian topological order. The gauge symmetry is enlarged to be $U(1) \rtimes \mathbb{Z}_2 = O(2)$, i.e. the theory becomes $O(2)_{2k}$ Chern-Simons theory. The new particle content is

- Charge $Q \neq 0, k$: these anyons become non-Abelian anyons with quantum dimension two, given by $Q \oplus (-Q)$.

- Charge $Q = 0$: the vacuum anyon and the $\mathbb{Z}_2$ symmetry defects lead to four anyons: the trivial anyon, the Abelian boson $W$ given by the $\mathbb{Z}_2$ Wilson line, and anyons $\xi_0, \chi_0$ of spin $\frac{1}{16}$ and $\frac{9}{16}$ respectively. The quantum dimension of $\xi_0$ and $\chi_0$ is $\sqrt{k}$. Therefore, these are non-Abelian for $k > 1$.

- Charge $Q = k$: this anyon and $\mathbb{Z}_2$ symmetry defects together with this anyon also lead to four anyons: the Abelian anyons $a_1, a_2$ with spin $\frac{k}{4}$, and anyons $\xi_k, \chi_k$ of spin $\frac{1}{16}$ and $\frac{9}{16}$ respectively. The quantum dimension of $\xi_k, \chi_k$ is also $\sqrt{k}$, so these are also non-Abelian for $k > 1$.

In the new theory $O(2)_{2k}$ the only Abelian anyons have spin $0, k/4$ mod 1, so one would not be able to detect the fractional quantum Hall conductance $\sigma_H = 1/(2k)$ for $k > 1$ from looking at the spins of the abelian anyons. It is instead related to the spin of the non-Abelian anyon $v \oplus v^{-1}$. Thus the non-invertible cosine symmetry $O(2)/\mathbb{Z}_2$ gives rise to a Hall conductance proportional to the spin of non-Abelian anyon, rather than an abelain one.

We remark that when $k = 1$, $O(2)_2 \leftrightarrow U(1)_8$ is an Abelian TQFT [56, 57]. Thus even in Abelian TQFTs there can be fractionalized non-invertible coset symmetry.

### 3.3.2 Gapless edge modes

Here we consider a $(1+1)$D edge state of the above FQH state with cosine symmetry. We argue that the nonzero electric Hall conductance (3.20) enforces a gapless edge mode if the boundary preserves the cosine symmetry. Before giving a proof for this statement, we first need to clarify what we mean by a symmetry-preserving edge state, since there can be multiple definitions for it which bifurcate for non-invertible symmetries [58]. In this paper, symmetry-preserving boundary condition means a boundary condition that satisfies the following two requirements:[6]

1. The boundary state is invariant the symmetry action. That is, the boundary state $|\mathcal{B}\rangle$ supported on a closed 2d space is an eigenstate of the symmetry operators $\mathcal{D}$: $\mathcal{D}|\mathcal{B}\rangle \propto |\mathcal{B}\rangle$. This is equivalent to requiring that the type of the boundary condition is invariant under pushing the symmetry defect onto the boundary.

2. The symmetry defect can terminate at the boundary. Its endpoint is topological and defines the symmetry defect at the boundary theory.

Let us show that the cosine symmetry-preserving boundary must carry $\sigma_H = 0$.

We take a symmetry-preserving gapped boundary state $|\mathcal{B}\rangle$. First, the invariance of the boundary state on a torus $|\mathcal{B}\rangle$ under the cosine symmetry defect $\tilde{U}_{\theta=0}$ implies that the Wilson line $W$ of the $\mathbb{Z}_2$ charge conjugation symmetry must be condensed at the boundary. This

---

[6]Following the terminology of [58], this amounts to requiring that the boundary condition is both strongly and weakly symmetric.

can be seen by noticing that $\tilde{U}_{\theta=0}$ is a condensation defect of $W$ due to (3.14), which acts on a torus $T_{xy}^2$ by an operator $(1 + W(\gamma_x) + W(\gamma_y) + W(\gamma_x)W(\gamma_y))$.

Meanwhile, the boundary state $|\mathcal{B}\rangle$ on a torus contains a state $|1\rangle$ labeled by a trivial anyon 1, and has the form of $|\mathcal{B}\rangle = |1\rangle + Z_W |W\rangle + ...$ with some non-negative integer $Z_W$. One can then immediately see that the state $\tilde{U}_{\theta=0} |\mathcal{B}\rangle$ contains the state $|W\rangle$ with some positive coefficient, which implies that $W$ is condensed at the boundary once we require $\tilde{U}_{\theta=0} |\mathcal{B}\rangle \propto |\mathcal{B}\rangle$.

Therefore, it must be possible to obtain the gapped boundary must by first condensing $W$ in the theory $\mathcal{T}/\mathbb{Z}_2$. This brings the theory back to an original FQH state $\mathcal{T}$ with $U(1) \rtimes \mathbb{Z}_2$ symmetry. If there is a gapped boundary, then we can further condense Lagrangian algebra anyons in $\mathcal{T}$. In other words, without loss of generality, we can describe the boundary state $|\mathcal{B}\rangle$ of $\mathcal{T}/\mathbb{Z}_2$ by first acting the operator $\mathcal{D}_{\text{Rep}(\mathbb{Z}_2)}$ on some gapped boundary state $|\mathcal{B}'\rangle$ of the theory $\mathcal{T}$,

$$|\mathcal{B}\rangle = \mathcal{D}_{\text{Rep}(\mathbb{Z}_2)} |\mathcal{B}'\rangle \qquad (3.21)$$

where $\mathcal{D}_{\text{Rep}(\mathbb{Z}_2)}$ is an interface operator obtained by half-gauging (see section 3.2 for discussions). The gapped boundary $\mathcal{B}$ is obtained by pushing the defect $\mathcal{D}_{\text{Rep}(\mathbb{Z}_2)}$ in parallel onto the gapped boundary $\mathcal{B}'$.

Now let us consider a cosine symmetry defect $\tilde{U}_\theta$ terminating at the gapped boundary $\mathcal{B}$. Since the $\tilde{U}_\theta$ is transformed into $U_\theta$ or $U_{-\theta}$ by crossing through the interface $\mathcal{D}_{\text{Rep}(\mathbb{Z}_2)}$, one can see that the $U(1)$ defects $U_\theta$ have to end at the gapped boundary $\mathcal{B}'$, meaning that the gapped boundary $\mathcal{B}'$ preserves the $U(1)$ symmetry. This enforces $\sigma_H = 0$.

## 3.4 More example of fractionalized coset symmetry: $S_3$ gauge theory

Let us consider $\mathbb{Z}_3$ gauge theory enriched by $U(1)$ symmetry, where the fractionalization is given by braiding with $v = em$. Since $v$ carries the spin $h_v = 1/3$, this topological order has the Hall response $\sigma_H = 2/3$. If we gauge the charge conjugation symmetry, we get $S_3$ gauge theory with the Hall response for the cosine symmetry.

There are eight anyons in the $S_3$ gauge theory (see e.g. [59]), which are usually denoted by $A, B, C, D, E, F, G, H$. Each anyon is associated with a conjugacy class of $S_3$ and an irreducible representation of the centralizer of a representative element from the conjugacy class:

- $\{1\}$: the centralizer is $S_3$ with the trivial ($A$), sign ($B$), and two dimensional ($C$) irreps.

- $\{s, sr, sr^2\}$: the centralizer is $\{1, s\}$ with the trivial ($D$) and sign ($E$) irreps.

- $\{r, r^2\}$: the centralizer is $\{1, r, r^2\}$ with the trivial ($F$), $e^{2\pi i/3}$ ($G$) and $e^{4\pi i/3}$ ($G$) irreps.

The $T$ matrix is $\text{diag}(1, 1, 1, -1, 1, 1, e^{2\pi i/3}, e^{4\pi i/3})$. $A$ and $B$ correspond to the vacuum of the $\mathbb{Z}_3$ gauge theory with and without the charge conjugation charge respectively. $C$ corresponds to the orbit $(e, e^2)$ under charge conjugation. $D$ and $E$ correspond to $\mathbb{Z}_2$ charge conjugation defects with and without the charge conjugation charge. $F, G$, and $H$ correspond to the orbits $(m, m^2), (em, e^2m^2)$, and $(e^2m, em^2)$. Note that the $S_3$ gauge theory does not have an Abelian anyon with the spin $1/3$, so the Hall conductance $\sigma_H = 2/3$ cannot be accounted for by ordinary fractionalization using Abelian anyons.

Based on the example discussed in section 3.4, we can describe the cosine symmetry defect of $S_3$ gauge theory as a sandwich of gapped interfaces of $S_3$ gauge theory. As discussed in section 3.2, this perspective allows us to describe the junction of the symmetry defects of cosine symmetry. The junction of the cosine symmetry defects that corresponds to fusion outcome $\tilde{U}_{[\theta]} \times \tilde{U}_{[\theta']} \to \tilde{U}_{[\theta+\theta']}$ can be realized by the junction of invertible symmetry $U(1)$ defects among $\theta, \theta', \theta + \theta'$ decorated with non-invertible defects, see Figure 3. Similarly, the junction for the fusion outcome $\tilde{U}_{[\theta]} \times \tilde{U}_{[\theta']} \to \tilde{U}_{[\theta-\theta']}$ can also be obtained from the junction of invertible symmetry $U(1)$ defects among $\theta, -\theta', \theta - \theta'$.

The symmetry fractionalization of a non-Abelian anyon $G = (em, e^2m^2)$ in $S_3$ gauge theory is illustrated in Figure 3. When the line operator of the non-Abelian anyon $G$ crosses through the junction of cosine symmetries, the junction non-trivially acts on the anyon $G$, which is regarded as the symmetry fractionalization.

Unlike the standard fractionalization of the invertible symmetry, the fractional charge of the cosine symmetry is given by a superposition of opposite $U(1)$ fractional charges. It reflects Figure 3: the fractional charge depends on whether it splits into $em$ or $e^2m^2$ with the opposite $U(1)$ charge when the $S_3$ gauge theory is condensed into a $\mathbb{Z}_3$ gauge theory.

# 4 Lattice Model with Fractionalized Non-Invertible Symmetry

In this section, we demonstrate the properties of the non-invertible symmetry fractionalization with a microscopic lattice model. We consider the (2+1)D quantum double model on a square lattice, which is a standard lattice model effectively described by a finite $G$ gauge theory. We find that the quantum double model with the gauge group $G = S_3$ has a fractionalized cosine symmetry.

## 4.1 Cosine symmetry in $S_3$ quantum double model

**Warm-up: $\mathbb{Z}_3$ gauge theory with $U(1)$ symmetry** Let us start with $\mathbb{Z}_3$ toric code with $U(1)$ symmetry in (2+1)D. We consider a square lattice with a $\mathbb{Z}_3$ qudit on each edge, with

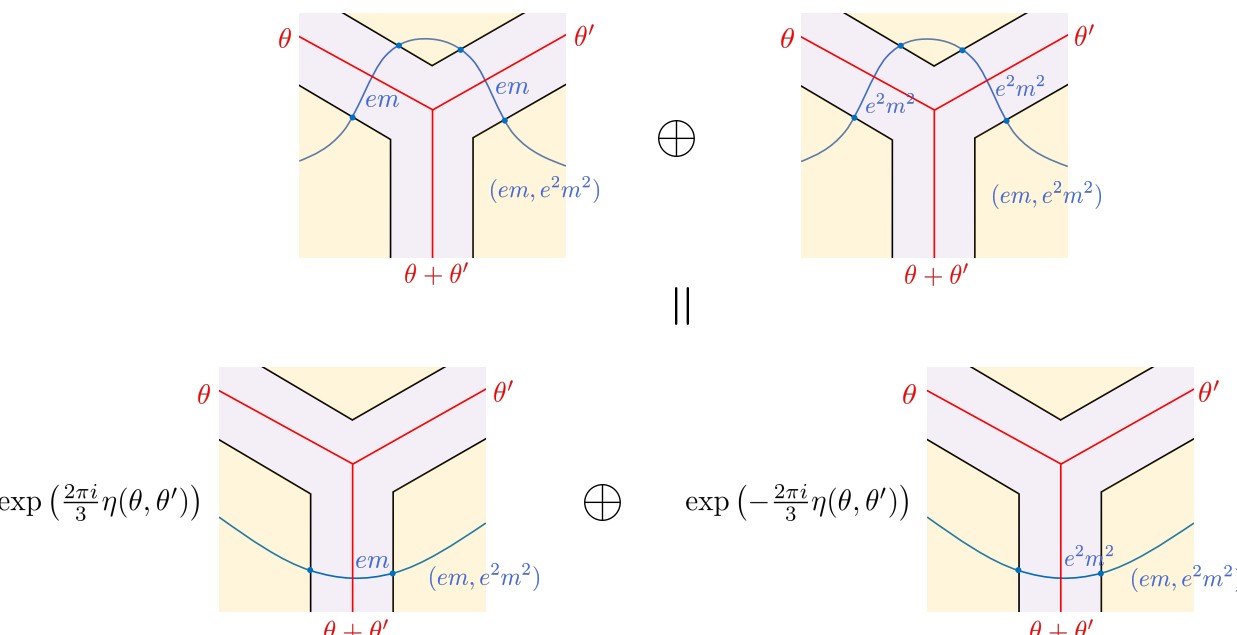

Figure 3: The symmetry fractionalizaton of the non-Abelian anyon $G = (em, e^2m^2)$. This anyon splits into $em$, $e^2m^2$ of $\mathbb{Z}_3$ gauge theory and they carry the opposite fractional charge. As a result, the fractional charge carried by $G$ is a diagonal matrix $\text{diag}(1/3, 2/3)$ rather than a number.

the Hamiltonian

$$H_{\mathbb{Z}_3} = -\sum_v A_v - \sum_p B_p \tag{4.1}$$

where $v$ and $p$ denote the vertex and plaquette respectively. Specifically, each term is given by

$$A_v = \frac{1}{3}\left(1 + X_{N(v)}X_{E(v)}X_{W(v)}^\dagger X_{S(v)}^\dagger + X_{N(v)}^\dagger X_{E(v)}^\dagger X_{W(v)}X_{S(v)}\right) , \tag{4.2}$$

$$B_p = \frac{1}{3}\left(1 + Z_{01}Z_{13}Z_{02}^\dagger Z_{23}^\dagger + Z_{01}^\dagger Z_{13}^\dagger Z_{02}Z_{23}\right) . \tag{4.3}$$

These two terms are illustrated in Figure 4.

We define the $\mathbb{Z}_3$ gauge field $a$ on links satisfying $e^{\frac{2\pi i}{3}a_e} = Z_e$. The state with $B_p = 1$ corresponds to the flat $\mathbb{Z}_3$ gauge field $da = 0$. $A_v = 1$ corresponds to the Gauss law constraint of the $\mathbb{Z}_3$ gauge field, so the ground state Hilbert space is described by the $\mathbb{Z}_3$ gauge theory.

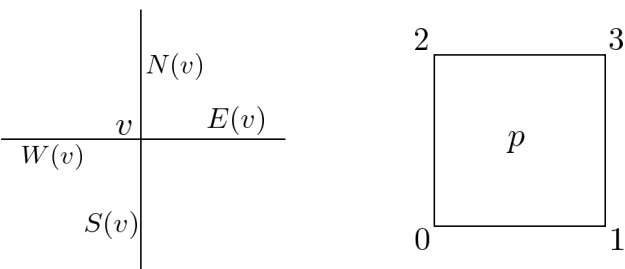

Figure 4: The edges nearby a vertex $v$ and plaquette $p$.

The $U(1)$ global symmetry $U_\theta$ is defined as the action on the state

$$U_\theta \ket{a} = \exp\left(i[\theta]_{2\pi} \int \frac{d\hat{a}}{3}\right) \ket{a}. \tag{4.4}$$

where $\hat{a}$ is the integral lift of $a$. This $U(1)$ symmetry exhibits the symmetry fractionalization. When we write the electric particle of the $\mathbb{Z}_3$ gauge theory as labeled by $e$, the Abelian anyon $e^{\eta(\theta_1,\theta_2)}$ is decorated at the junction of $U(1)$ transformations $\theta_1, \theta_2$, and $\theta_1 + \theta_2$ mod $2\pi$.

**Cosine symmetry of $S_3$ gauge theory**   We then consider the $S_3$ quantum double model in (2+1)D. The local Hilbert space on each edge has 6 dimensions, whose basis states $\{\ket{g}\}$ are labeled by group elements $g \in S_3$. The Hamiltonian is given by

$$H_{S_3} = -\sum_v A_v - \sum_p B_p \tag{4.5}$$

with each term given by

$$A_v = \frac{1}{|G|} \sum_{g \in G} \overrightarrow{X}^g_{N(v)} \overrightarrow{X}^g_{E(v)} \overleftarrow{X}^{g^{-1}}_{W(v)} \overleftarrow{X}^{g^{-1}}_{S(v)}, \quad B_p = \delta_{g_{01}g_{13}g_{02}^{-1}g_{23}^{-1},0}. \tag{4.6}$$

Here we defined the Pauli $X$ like operators as

$$\overrightarrow{X}^g \ket{h} = \ket{gh}, \quad \overleftarrow{X}^{g^{-1}} \ket{h} = \ket{hg^{-1}}. \tag{4.7}$$

It is convenient to label $g \in S_3$ by a pair $g = (a, b)$ with $a \in \mathbb{Z}_3, b \in \mathbb{Z}_2$, satisfying the group multiplication law

$$(a_1, b_1) \times (a_2, b_2) = (a_1 + (-1)^{b_1} a_2, b_1 + b_2). \tag{4.8}$$

$a, b$ are regarded as $\mathbb{Z}_3$ and $\mathbb{Z}_2$ gauge fields respectively.

Let us define the operators acting like Pauli $X, Z$ operators on the $a, b$ fields:

$$Z^a \ket{a, b} = e^{\frac{2\pi i}{3}a} \ket{a, b}, \quad X^a \ket{a, b} = \ket{a + 1, b} \tag{4.9}$$

$$Z^b \ket{a,b} = (-1)^b \ket{a,b}, \ \overrightarrow{X}^b \ket{a,b} = \ket{-a, b+1}, \ \overleftarrow{X}^b \ket{a,b} = \ket{a, b+1} \tag{4.10}$$

$\overrightarrow{X}^b, \overleftarrow{X}^b$ corresponds to the left and right action of $(0,1) \in S_3$ on $(a,b)$ respectively.

The vertex term of the $S_3$ quantum double model can then be expressed as

$$A_v = A_v^a A_v^b \tag{4.11}$$

where $A_v^a$ is the vertex term of the $\mathbb{Z}_3$ toric code expressed by $X^a$, and

$$A_v^b = \frac{1 + \overrightarrow{X}_{N(v)}^b \overrightarrow{X}_{E(v)}^b \overleftarrow{X}_{W(v)}^b \overleftarrow{X}_{S(v)}^b}{2} \tag{4.12}$$

We then define the projection operator

$$\mathcal{D}^b = \prod_e \left( \frac{1 + Z_e^b}{2} \right), \tag{4.13}$$

and the projection operator

$$\Pi^b = \prod_v A_v^b \tag{4.14}$$

Now we are ready to describe the cosine symmetry of the $S_3$ quantum double model (4.5). The generator of the cosine symmetry is given by

$$\tilde{U}_\theta = 2^{|v|} \cdot \Pi^b U_\theta \mathcal{D}^b \tag{4.15}$$

where $|v|$ is the number of vertices. This generates an emergent non-invertible symmetry of $S_3$ gauge theory within the low energy subspace where $\Pi^b = 1$ is satisfied. Within this subspace, $\tilde{U}_\theta$ is expressed as

$$\tilde{U}_\theta = 2^{|v|} \cdot \Pi^b \mathcal{D}^b U_\theta \mathcal{D}^b \Pi^b . \tag{4.16}$$

This operator (4.16) generates an exact symmetry of the $S_3$ quantum double model. Note that since we have a projector $\Pi^b$ on the right, the operator (4.16) annihilates the state with electric particle of $\mathbb{Z}_2$ charge conjugation symmetry, while (4.15) does not.

In the expression (4.15), $\mathcal{D}^b$ condenses the Wilson line for the gauged charge conjugation symmetry, resulting in a $\mathbb{Z}_3$ gauge theory. $U_\theta$ is an invertible symmetry in $\mathbb{Z}_3$ gauge theory, and $\Pi^b$ then brings the theory back to $S_3$ gauge theory. Due to the projector $\mathcal{D}^b$, it eliminates the electric particle excitation of charge conjugation symmetry from the state. Since the particle excitations carry finite energy, its elimination implies that the operator (4.15) only commutes with the Hamiltonian in the subspace without $\mathbb{Z}_2$ electric particles, i.e., $\Pi^b = 1$.

We note that the form of the operator $\tilde{U}_\theta$ is aligned with the sandwich form (3.7); the operators $\Pi^b$ on the left, $\mathcal{D}^b$ on the right of (4.15) correspond to half-gauging defects $D_{\mathrm{Rep}(\mathbb{Z}_2)}$, $\overline{D}_{\mathrm{Rep}(\mathbb{Z}_2)}$ at a time slice. For the purpose of computing the fusion rule of operators below, we consider the low energy subspace where $\Pi^b = 1$ is satisfied. This allows us to derive the fusion rules for the coset symmetry in the absence of electric particle excitations for charge conjugation symmetry. If there are such excitations, the fusion rules can be modified to reflect the fractionalization of the symmetry on the excitations. We note that for invertible symmetries, the symmetry action on a pair of conjugate excitations is the same as the action on the vacuum, since the projective phases cancel. Here, when the symmetry is non-invertible, we need to be more careful about the distinction.

Below let us derive the properties of the operator:

- When $\theta = 0$,

$$\tilde{U}_{\theta=0} = 2^{|v|} \cdot \Pi^b \mathcal{D}^b \Pi^b \tag{4.17}$$

Here, note that $\mathcal{D}^b$ can be rewritten as the sum of open string operators

$$\mathcal{D}^b = \frac{1}{2^{|e|}} \sum_{C \in C_1(M, \mathbb{Z}_2)} \left( \prod_{e \subset C} Z_e^b \right) \tag{4.18}$$

where the sum of $C$ is over all possible 1-chains of the square lattice, and $|e|$ is the number of edges. When projected onto the states with $A_v^b = 1$, only the closed strings $C$ survives, since the product of $Z_e^b$ is a line operator of the Abelian anyon in $S_3$ gauge theory and it excites $A_v^b$ at its ends. We then have

$$\tilde{U}_{\theta=0} = \Pi^b \left( 2^{\chi-1} \sum_{C \in H^1(M, \mathbb{Z}_2)} \left( \prod_{e \in C} Z_e^b \right) \right) \Pi^b, \tag{4.19}$$

where $\chi = |v| - |e| + |p|$ is the Euler characteristic of the lattice. This is a condensation operator for the Abelian boson of $S_3$ gauge theory that is the Wilson line from gauging the charge conjugation symmetry. See Refs. [29, 60] for other examples of condensation operators on (2+1)D lattice models.

- Let us then derive the fusion rules of the operators. One can derive

$$\tilde{U}_\theta \times \tilde{U}_{\theta'} = \tilde{U}_{\theta+\theta'} + \tilde{U}_{\theta-\theta'} \tag{4.20}$$

To see this, it is convenient to find a simple expression of the operator $\mathcal{D}^b \Pi^b \mathcal{D}^b$. Since the operator $\overrightarrow{X}_{N(v)}^b \overrightarrow{X}_{E(v)}^b \overleftarrow{X}_{W(v)}^b \overleftarrow{X}_{S(v)}^b$ shifts $b$ fields $b \to b + d\hat{v}$, most of the product of these vertex terms are projected out by $\mathcal{D}^b$. We can get

$$\mathcal{D}^b \Pi^b \mathcal{D}^b = \mathcal{D}^b \left( \prod_v A_v^b \right) \mathcal{D}^b = \frac{1}{2^{|v|}} \mathcal{D}^b \left( 1 + V_{\mathbb{Z}_2}^C \right) \mathcal{D}^b \tag{4.21}$$

with

$$V_{\mathbb{Z}_2}^C = \prod_v \overrightarrow{X}^b_{N(v)} \overrightarrow{X}^b_{E(v)} \overleftarrow{X}^b_{W(v)} \overleftarrow{X}^b_{S(v)} \tag{4.22}$$

One can see that $V_{\mathbb{Z}_2}^C$ leaves $b$ fields invariant, while it acts by charge conjugation on the $\mathbb{Z}_3$ gauge field $a$,

$$V_{\mathbb{Z}_2}^C |\{a\}\rangle = |\{-a\}\rangle . \tag{4.23}$$

The fusion rule of $\tilde{U}_\theta$ is derived by

$$
\begin{aligned}
\tilde{U}_\theta \times \tilde{U}_{\theta'} &= 2^{|2v|} \cdot \Pi^b \mathcal{D}^b U_\theta \mathcal{D}^b \Pi^b \mathcal{D}^b U_{\theta'} \mathcal{D}^b \Pi^b \\
&= 2^{|v|} \cdot \Pi^b \mathcal{D}^b U_\theta \mathcal{D}^b \left(1 + V_{\mathbb{Z}_2}^C\right) \mathcal{D}^b U_{\theta'} \mathcal{D}^b \Pi^b \\
&= 2^{|v|} \cdot \Pi^b \mathcal{D}^b U_\theta \left(1 + V_{\mathbb{Z}_2}^C\right) U_{\theta'} \mathcal{D}^b \Pi^b \\
&= 2^{|v|} \cdot \Pi^b \mathcal{D}^b U_\theta \left(U_{\theta'} + U_{-\theta'} V_{\mathbb{Z}_2}^C\right) \mathcal{D}^b \Pi^b \\
&= 2^{|v|} \cdot \Pi^b \mathcal{D}^b U_{\theta+\theta'} \mathcal{D}^b \Pi^b + 2^{|v|} \cdot \Pi^b \mathcal{D}^b U_{\theta-\theta'} V_{\mathbb{Z}_2}^C \mathcal{D}^b \Pi^b \\
&= 2^{|v|} \cdot \Pi^b \mathcal{D}^b U_{\theta+\theta'} \mathcal{D}^b \Pi^b + 2^{|v|} \cdot \Pi^b \mathcal{D}^b U_{\theta-\theta'} \mathcal{D}^b \Pi^b \\
&= \tilde{U}_{\theta+\theta'} + \tilde{U}_{\theta-\theta'}
\end{aligned}
\tag{4.24}
$$

## 4.2 Fractionalization of cosine symmetry in $S_3$ quantum double model

**Warm-up: Fractionalization of $U(1)$ symmetry in $\mathbb{Z}_3$ gauge theory**   Let us consider a state with anyon excitations, and suppose that an anyon $x$ is separated from other excitations. Let us pick a subsystem $R$ that contains the $x$ excitation. The fusion rule of the $U(1)$ symmetry generators support at $R$ is given by

$$U_\theta(R) \times U_{\theta'}(R) = U_{\theta+\theta'}(R) \times W_e(\partial R)^{\eta(\theta,\theta')}, \tag{4.25}$$

where $\eta(\theta,\theta') = \frac{[\theta]_{2\pi} + [\theta']_{2\pi} - [\theta+\theta']_{2\pi}}{2\pi}$. Here, $W_e(\partial R)^{\eta(\theta,\theta')}$ is a closed string operator for $e^{\eta(\theta,\theta')}$, that wraps around the boundary of $R$. Note that when the closed string operator acts on a state with no anyon excitations, it gets absorbed into the ground state. However, when there is an anyon excitation in $R$, the string operator produces a braiding phase between the anyon and $e^{\eta(\theta,\theta')}$. This implies that the difference between the action of $U_\theta \times U_{\theta'}$ and $U_{\theta+\theta'}$ on the anyon $x$ is given by the fractional phase

$$
\begin{aligned}
U_\theta(R) \times U_{\theta'}(R) |x,\ldots\rangle &= U_{\theta+\theta'}(R) \times W_e(\partial R)^{\eta(\theta,\theta')} |x,\ldots\rangle \\
&= U_{\theta+\theta'}(R) \times \exp\left(2\pi i q_x \eta(\theta,\theta')\right) |x,\ldots\rangle
\end{aligned}
\tag{4.26}
$$

where $\exp(2\pi i q_x) = M_{x,e}$, and $M_{x,e}$ denotes the mutual braiding between two particles $x, e$. The phase proportional to the fractional charge $q_x$ manifests the symmetry fractionalization on the anyon $x$.

**Fractionalization of cosine symmetry in $S_3$ gauge theory**  We will see that the cosine symmetry $\tilde{U}_\theta$ induces the symmetry fractionalization of the anyon $(m, m^2)$ in the $S_3$ gauge theory. The anyon $(m, m^2)$ is created by an open string operator on the line $\hat{\gamma}_{p,p'}$ terminating at the plaquettes $p, p'$. Given that each anyon carries the quantum dimension 2, the dimension of the Hilbert space with a pair of anyon excitations has 4 dimensions and is spanned by the states $|(j, k)\rangle$. Note that this 4 dimensional Hilbert space is technically not physical because the individual states cannot be distinguished by gauge invariant operators; $j, k \in \{1, 2\}$ labels the internal basis states of non-Abelian anyon at $p, p'$ respectively. The symmetry fractionalization can be seen by acting the symmetry generators at the region $R$ that contains the plaquette $p$ but not $p'$ (see Figure 6). We explicitly compute the action of cosine symmetry in the presence of anyon excitations, and show that

$$
\tilde{U}_\theta(R) \times \tilde{U}_{\theta'}(R) |(j, k)\rangle
$$
$$
= \left[\cos\left(\frac{2\pi}{3}\eta(\theta, \theta') + \frac{1}{3}[\theta + \theta']_{2\pi}\right) + \cos\left(\frac{2\pi}{3}\eta(\theta, -\theta') + \frac{1}{3}[\theta - \theta']_{2\pi}\right)\right] (|(1, k)\rangle + |(2, k)\rangle)
$$

(4.27)

and

$$
\left[\tilde{U}_{\theta+\theta'}(R) + \tilde{U}_{\theta-\theta'}(R)\right] |(j, k)\rangle
$$
$$
= \left[\cos\left(\frac{1}{3}[\theta + \theta']_{2\pi}\right) + \cos\left(\frac{1}{3}[\theta - \theta']_{2\pi}\right)\right] (|(1, k)\rangle + |(2, k)\rangle)
$$

(4.28)

By comparing the above two expressions, one can see that the fusion rule $\tilde{U}_\theta \times \tilde{U}_{\theta'} = \tilde{U}_{\theta+\theta'} + \tilde{U}_{\theta-\theta'}$ is modified by the fractional charge proportional to $\frac{2\pi}{3}\eta(\theta, \pm\theta')$. This implies that the non-invertible cosine symmetry is fractionalized on a non-Abelian anyon $(m, m^2)$.

There are two points special about fractionalization of cosine symmetry:

- The fusion of cosine symmetries $\tilde{U}_\theta \times \tilde{U}_{\theta'}$ splits into two fusion outcomes $\tilde{U}_{\theta+\theta'}$ or $\tilde{U}_{\theta-\theta'}$. Depending on the fusion outcome, the effect of symmetry fractionalization becomes different; in the former case it depends on $\eta(\theta, \theta')$, while in the latter $\eta(\theta, -\theta')$.

- Once we focus on one fusion outcome $\tilde{U}_{\theta\pm\theta'}$, the symmetry fractionalization appears as the difference between $\tilde{U}_\theta \times \tilde{U}_{\theta'}$ and $\tilde{U}_{\theta\pm\theta'}$ in the cosine of the fractional $U(1)$ charge carried by an anyon.[7]

Below, we confirm (4.27), (4.28). First, we have

$$
\tilde{U}_\theta(R) \times \tilde{U}_{\theta'}(R) = 2^{|v|} \cdot \left[\Pi^b \mathcal{D}^b U_\theta U_{\theta'} \mathcal{D}^b \Pi^b\right]_R + 2^{|v|} \cdot \left[\Pi^b \mathcal{D}^b U_\theta U_{-\theta'} \mathcal{D}^b \Pi^b\right]_R
$$

(4.29)

---

[7]We note that cosine symmetry acts on the state $|(1, k)\rangle + |(2, k)\rangle$ by an overall normalization given by cosine of fractional $U(1)$ charge. In particular, the above fusion rule can annihilate the states with specific choice of $\theta, \theta'$. One could normalize the state $|(1, k)\rangle + |(2, k)\rangle$ to suppress the effect of fractionalization with specific $\theta, \theta'$, but one cannot choose a normalization which eliminates the fractionalization of all symmetry operators.

See Figure 5 for details of the operators $\Pi^b, \mathcal{D}^b, U_\theta$ defined on the region $R$.

Let us describe an open string operator for the anyon $(m, m^2)$ in $S_3$ gauge theory. We consider a straight line of the dual lattice $\hat{\gamma}_{p,p'}$ with length $L$ terminating at two plaquettes $p$, $p'$. Then, the string operator $W^{(j,k)}_{(m,m^2)}(\hat{\gamma}_{p,p'})$ is labeled by $(j, k)$, where $j, k \in \{1, 2\}$ labels the basis for the Hilbert space with anyons with quantum dimension 2. The Hilbert with a pair of anyons $(m, m^2)$ has 4 dimensions, and its basis is given by $|(j, k)\rangle = W^{(j,k)}_{(m,m^2)}(\hat{\gamma}_{p,p'})|\text{GS}\rangle$.

The string operator is then given by [50]

$$
\begin{aligned}
W^{(1,1)}_{(m,m^2)}(\hat{\gamma}_{p,p'}) &= \left(\prod_{j=0}^{L}(X^a_{\hat{e}_j})^{\Pi_{k=0}^{j-1} Z^b_{e_k}}\right) \times \frac{1 + \prod_{k=0}^{L-1} Z^b_{e_k}}{\sqrt{2}} \\
W^{(1,2)}_{(m,m^2)}(\hat{\gamma}_{p,p'}) &= \left(\prod_{j=0}^{L}(X^a_{\hat{e}_j})^{\Pi_{k=0}^{j-1} Z^b_{e_k}}\right) \times \frac{1 - \prod_{k=0}^{L-1} Z^b_{e_k}}{\sqrt{2}} \\
W^{(2,1)}_{(m,m^2)}(\hat{\gamma}_{p,p'}) &= \left(\prod_{j=0}^{L}(X^{a\dagger}_{\hat{e}_j})^{\Pi_{k=0}^{j-1} Z^b_{e_k}}\right) \times \frac{1 - \prod_{k=0}^{L-1} Z^b_{e_k}}{\sqrt{2}} \\
W^{(2,2)}_{(m,m^2)}(\hat{\gamma}_{p,p'}) &= \left(\prod_{j=0}^{L}(X^{a\dagger}_{\hat{e}_j})^{\Pi_{k=0}^{j-1} Z^b_{e_k}}\right) \times \frac{1 + \prod_{k=0}^{L-1} Z^b_{e_k}}{\sqrt{2}}
\end{aligned}
\tag{4.30}
$$

where we label the edges by numbers starting with 0 at the plaquette $p$, and terminating with $L$ at the plaquette $p'$. $\hat{e}_j$ is an edge of $\hat{\gamma}_{p,p'}$ cutting the edges, and $e_k$ is an edge of $\gamma_{p,p'}$. See Figure 6 (b) for an illustration.

Let us consider a line operator $W^{(j,k)}_{(m,m^2)}(\hat{\gamma}_{p,p'})$ starting at a plaquette $p$ inside the region $R$. The termination $p'$ is away from the region $R$. The state with an anyon $(m, m^2)$ at the plaquette $p$ is given by acting any of the four line operators $W^{(j,k)}_{(m,m^2)}(\hat{\gamma}_{p,p'})$ on the ground state. See Figure 6 (a) for the geometry.

The symmetry fractionalization on $(m, m^2)$ can be seen by acting symmetry generators $\tilde{U}_\theta(R)$ in the presence of the anyon.

$$
\begin{aligned}
\tilde{U}_\theta(R) \times \tilde{U}_{\theta'}(R)|(j,k)\rangle = {}& 2^{|v|} \cdot \left[\Pi^b \mathcal{D}^b U_\theta U_{\theta'} \mathcal{D}^b \Pi^b\right]_R W^{(j,k)}_{(m,m^2)}(\hat{\gamma}_{p,p'})|\text{GS}\rangle \\
&+ 2^{|v|} \cdot \left[\Pi^b \mathcal{D}^b U_\theta U_{-\theta'} \mathcal{D}^b \Pi^b\right]_R W^{(j,k)}_{(m,m^2)}(\hat{\gamma}_{p,p'})|\text{GS}\rangle
\end{aligned}
\tag{4.31}
$$

The projector $\Pi_b = \prod_v A_v$ behaves as an identity operator away from the anyon excitations, but $A_v$ can change the internal state of the anyon $|(j,k)\rangle$ when $v$ is located at the position of the anyon. More concretely, the star operator $\overrightarrow{X}^b_{N(v)} \overrightarrow{X}^b_{E(v)} \overleftarrow{X}^b_{W(v)} \overleftarrow{X}^b_{S(v)}$ acts by charge conjugation on the internal state of the non-Abelian anyon. One can explicitly check that

$$
\begin{aligned}
\left[\Pi^b\right]_R W^{(1,k)}_{(m,m^2)}(\hat{\gamma}_{p,p'})|\text{GS}\rangle &= \frac{1}{2}\left(W^{(1,k)}_{(m,m^2)}(\hat{\gamma}_{p,p'}) + W^{(2,k)}_{(m,m^2)}(\hat{\gamma}_{p,p'})\right)|\text{GS}\rangle \\
\left[\Pi^b\right]_R W^{(2,k)}_{(m,m^2)}(\hat{\gamma}_{p,p'})|\text{GS}\rangle &= \frac{1}{2}\left(W^{(1,k)}_{(m,m^2)}(\hat{\gamma}_{p,p'}) + W^{(2,k)}_{(m,m^2)}(\hat{\gamma}_{p,p'})\right)|\text{GS}\rangle
\end{aligned}
\tag{4.32}
$$

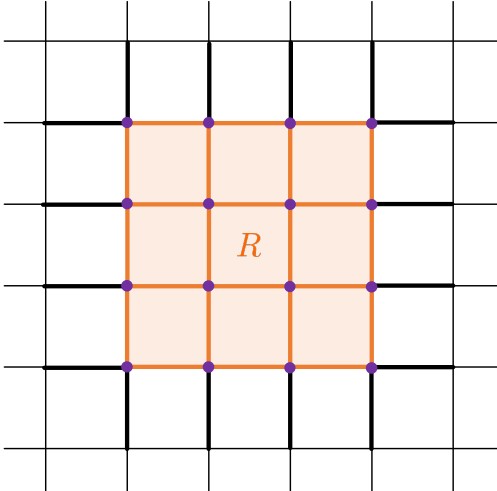

Figure 5: The region $R$ consists of a set of plaquettes inside a disk. The edges of the plaquettes in $R$ are represented by orange lines, and the vertices of plaquettes in $R$ are represented by purple dots. Then, $\Pi^b$ is the product of $A_v^b$ over the purple vertices $v$, $\mathcal{D}^b$ is the product of $(1 + Z^b)/2$ over the orange edges $e$, and $U_\theta$ is the integral $\exp\big(i[\theta]_{2\pi} \int d\hat{a}/3\big)$ over the orange region $R$. Note that $\Pi^b$ also acts on thick black edges, since the operators $A_v^b$ on the boundary of $R$ act on these vertices.

After the wave function is projected by the operator $[\mathcal{D}^b]_R$, the $\mathbb{Z}_2$ gauge field satisfies $Z^b = 1$ inside the region $R$. According to the expression in (4.30), the expression of string operator inside the region $R$ is fixed as

$$W^{(1,k)}_{(m,m^2)}(\hat{\gamma}_{p,p'}) = \prod_j X^a_{\hat{e}_j} \quad \text{when restricted to the region } R. \tag{4.33}$$

$$W^{(2,k)}_{(m,m^2)}(\hat{\gamma}_{p,p'}) = \prod_j X^{a\dagger}_{\hat{e}_j} \quad \text{when restricted to the region } R. \tag{4.34}$$

With this in mind, one can rewrite the expression as

$$
\begin{aligned}
&\tilde{U}_\theta(R) \times \tilde{U}_{\theta'}(R) \,|(j,k)\rangle \\
&= 2^{|v|} \cdot \left[\Pi^b \mathcal{D}^b U_\theta U_{\theta'} \mathcal{D}^b \Pi^b\right]_R \cdot \frac{1}{2}\left(W^{(1,k)}_{(m,m^2)}(\hat{\gamma}_{p,p'}) + W^{(2,k)}_{(m,m^2)}(\hat{\gamma}_{p,p'})\right) |\mathrm{GS}\rangle \\
&+ 2^{|v|} \cdot \left[\Pi^b \mathcal{D}^b U_\theta U_{-\theta'} \mathcal{D}^b \Pi^b\right]_R \frac{1}{2}\left(W^{(1,k)}_{(m,m^2)}(\hat{\gamma}_{p,p'}) + W^{(2,k)}_{(m,m^2)}(\hat{\gamma}_{p,p'})\right) |\mathrm{GS}\rangle \\
&= 2^{|v|} \cdot \left[\Pi^b \mathcal{D}^b\right]_R \sum_j \left[\frac{1}{2}\exp\left(\frac{ij}{3}([\theta]_{2\pi} + [\theta']_{2\pi})\right) W^{(j,k)}_{(m,m^2)}(\hat{\gamma}_{p,p'})\right] |\mathrm{GS}\rangle \\
&+ 2^{|v|} \cdot \left[\Pi^b \mathcal{D}^b\right]_R \sum_j \left[\frac{1}{2}\exp\left(\frac{ij}{3}([\theta]_{2\pi} + [-\theta']_{2\pi})\right) W^{(j,k)}_{(m,m^2)}(\hat{\gamma}_{p,p'})\right] |\mathrm{GS}\rangle
\end{aligned}
\tag{4.35}
$$

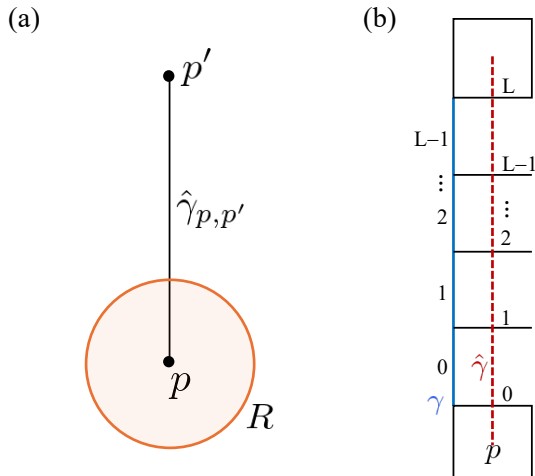

Figure 6: The anyon created by an open line operator $\hat{\gamma}_p$.

One can then see that

$$2^{|v|} \cdot \left[ \Pi^b \mathcal{D}^b \right]_R W_{(m,m^2)}^{(j,k)}(\hat{\gamma}_{p,p'}) \left| \text{GS} \right\rangle = \left( W_{(m,m^2)}^{(1,k)}(\hat{\gamma}_{p,p'}) + W_{(m,m^2)}^{(2,k)}(\hat{\gamma}_{p,p'}) \right) \left| \text{GS} \right\rangle \qquad (4.36)$$

so we have

$$\tilde{U}_\theta(R) \times \tilde{U}_{\theta'}(R) \left| (j,k) \right\rangle$$
$$= \cos\left( \frac{1}{3}([\theta]_{2\pi} + [\theta']_{2\pi}) \right) \left( |(1,k)\rangle + |(2,k)\rangle \right) + \cos\left( \frac{1}{3}([\theta]_{2\pi} + [-\theta']_{2\pi}) \right) \left( |(1,k)\rangle + |(2,k)\rangle \right)$$
$$\qquad (4.37)$$

which produces (4.27). Meanwhile, the action of $\tilde{U}_{\theta+\theta'}(R)$, $\tilde{U}_{\theta-\theta'}(R)$ are evaluated as

$$\tilde{U}_{\theta+\theta'}(R) \left| (j,k) \right\rangle = \cos\left( \frac{1}{3}[\theta + \theta']_{2\pi} \right) \left( |(1,k)\rangle + |(2,k)\rangle \right),$$
$$\tilde{U}_{\theta-\theta'}(R) \left| (j,k) \right\rangle = \cos\left( \frac{1}{3}[\theta - \theta']_{2\pi} \right) \left( |(1,k)\rangle + |(2,k)\rangle \right) \qquad (4.38)$$

which produces (4.28).

# 5 General Coset Symmetry Fractionalization and Bulk TQFT

## 5.1 General coset symmetry and its fractionalization in (2+1)D topological order

We now describe the general framework for describing fractionalization of non-invertible coset symmetry $G/K$ in (2+1)D topological orders. We start with a (2+1)D TQFT with 0-form

global symmetry $G$ and gauge the non-normal subgroup $K$. $G$ can either be a continuous or discrete group. The original TQFT is described by a modular tensor category $\mathcal{C}$, whose objects $\mathrm{Obj}(\mathcal{C})$ corresponds to the set of anyons. When $G$ is not a connected group, the $G$ symmetry can act on anyons by permuting their labels according to different connected components $\pi_0(G)$, $\rho_g : a \to {}^g a$ for $a \in \mathrm{Obj}(\mathcal{C}), g \in G$. The $G$ symmetry action on the TQFT is then characterized by the symmetry fractionalization data $\{U, \eta\}$ (see Figure 7).

Let us gauge the non-normal $K$ subgroup of the modular tensor category $\mathcal{C}$. The gauged theory is described by a $K$-equivalentization of the category $\mathcal{C}$, which we denote as $\mathcal{C}/K$. Algebraically, the gauging procedure to obtain $\mathcal{C}/K$ is performed in two steps [1]. The first step is to include the vortices of the symmetry group $K$, which is to take the $K$-crossed extension $\mathcal{C}_K^\times$ of the category $\mathcal{C}$,

$$\mathcal{C}_K^\times = \bigoplus_{k \in K} \mathcal{C}_k \tag{5.1}$$

where a simple object of $\mathcal{C}_k$ represents the vortex labeled by $k \in K$. The second step is to make the $K$ gauge group dynamical, and include the electric charge of $K$ symmetry. The simple anyon in the gauged theory is labeled by $([a_k], \pi_a) \in \mathrm{Obj}(\mathcal{C}/K)$, where $a_k \in \mathrm{Obj}(\mathcal{C}_k)$ is a vortex carrying the holonomy $k \in K$ in the ungauged theory, and $[a_k]$ denotes the orbit of $a_k$ under the permutation action of $K$. $\pi_a$ is an irreducible projective representation of the gauge group described as follows. Let us write a subgroup $K_a \subset K$ which fixes the label of the anyon $a_k$ under its action. $\pi_a$ then satisfies

$$\pi_a(k)\pi_a(k') = \eta_a(k, k')\pi_a(kk') \quad \text{for } k, k' \in K_a. \tag{5.2}$$

This is regarded as an electric charge attached to the particle. The above projective representation is referred to as an $\eta_a$-irrep, and its set is denoted by $\mathrm{Irrep}_\eta(K_a)$.

**Fractionalization of coset symmetry in the gauged theory $\mathcal{C}/K$** As described in section 3.2, the gauged theory $\mathcal{C}/K$ has the coset symmetry defect expressed by the sandwich

$$\tilde{U}_{[g]} = D_{\mathrm{Rep}(K)} \times U_g \times \overline{D}_{\mathrm{Rep}(K)}. \tag{5.3}$$

When the anyon $([a], \pi_a)$ of $\mathcal{C}/K$ tunnels through the non-invertible defect $\tilde{U}_{[g]}$, it can be transformed into multiple choices of anyons. Concretely, $\tilde{U}_{[g]}$ transforms the anyon according to the channels

$$([a], \pi_a) \to ([{}^{kgk^{-1}}a], \pi'_{kgk^{-1}{}_a}), \quad \text{for } k \in K, \pi'_{kgk^{-1}{}_a} \in \mathrm{Irrep}_\eta(K_{kgk^{-1}{}_a}) \tag{5.4}$$

Intuitively, the way the non-Abelian anyon $([a], \pi_a)$ is transformed by $\tilde{U}_{[g]}$ depends on the internal state of the non-Abelian anyon excitation. This leads to the multiple ways the anyon gets permuted (5.4), as well as the superposition for the distinct fractional charge of the anyon.

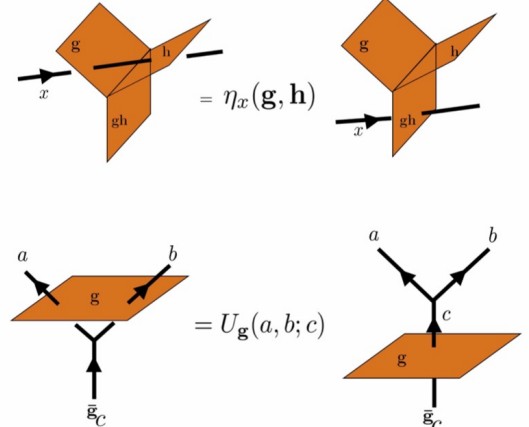

Figure 7: Anyon lines (black) passing through invertible symmetry defects $g, h \in G$ and graphical definitions of the $U$ and $\eta$ symbols.

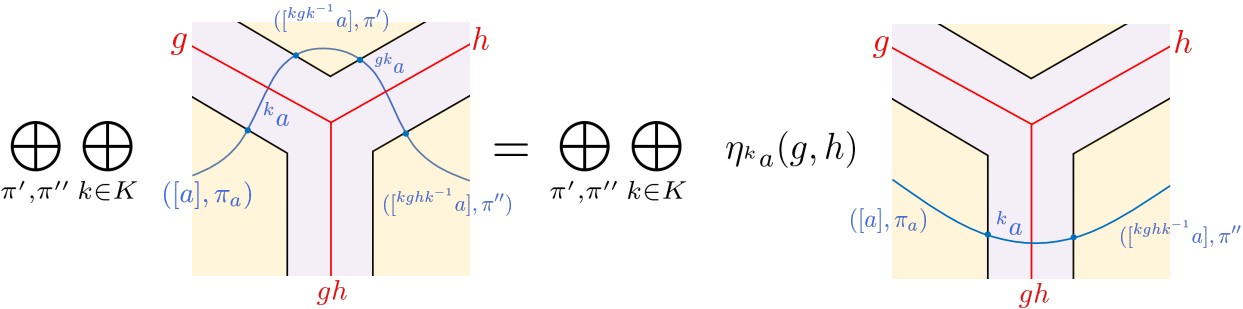

Figure 8: The anyon line $(a, \pi_a)$ crosses through the junction of the cosine symmetry defects. The fractional phase $\eta_{{}^k a}(g, h)$ appears depending on the way the anyon tunnels through the defect.

The pattern of the coset symmetry fractionalization depends on the choices of the transformation (5.4) labeled by $k \in K$. The effect of crossing the anyon through the junction of coset symmetry defects is described in Figure 8. The phase for the symmetry fractionalization $\eta_{{}^k a}(g, h)$ depends on the label of the anyon ${}^k a \in \mathrm{Obj}(\mathcal{C})$ inside the sandwich. This leads to the phenomena that the fractional charge is given by a superposition of distinct fractional phases, see section 3.4 for an example.

The action of the coset symmetry on the junction of anyons is described in Figure 9. Here, we introduced the tunneling matrix $M_{\mathcal{D}}^{\mu\nu}$ where the fusion vertex of the anyons $\mu$ is transformed into the new vertex $\nu$ by crossing through the gapped interface $\mathcal{D}$, see Figure 10. This quantity has been introduced in [61].

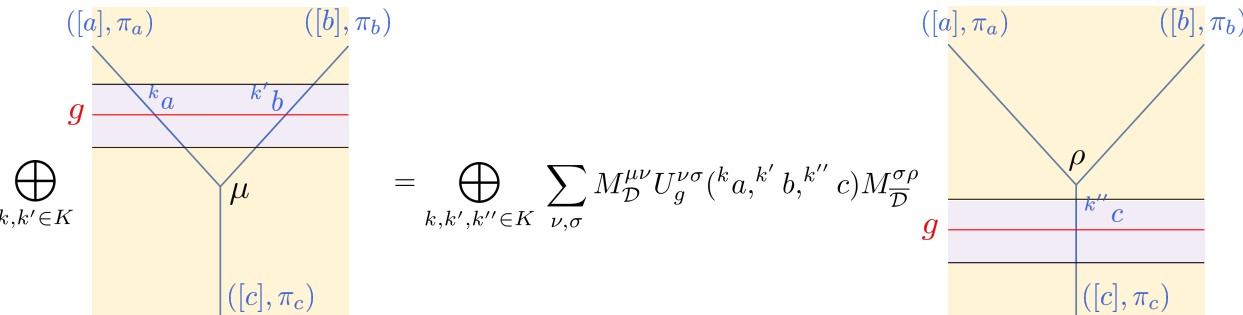

Figure 9: The cosine symmetry defect crosses through the junction of anyons. The phase factor $U$ appears depending on the way the anyon tunnels through the defect. $\mu, \nu, \sigma, \rho$ are the labels for the basis of the fusion vertex.

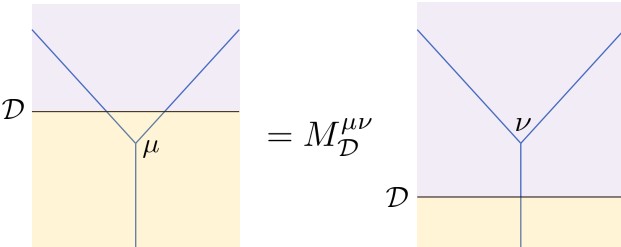

Figure 10: The tunneling matrix $M_{\mathcal{D}}$ is defined by the shift of correlation function when the gapped interface $\mathcal{D}$ crosses through the junction of anyons. For simplicity of figure, we have suppressed the label at the intersection between the anyon and $\mathcal{D}$.

**Redundancy in expressing the coset symmetry defect by a sandwich** As discussed in section 3.2, the sandwich of $g \in G$ or $kgk^{-1} \in G$ with $k \in K$ leads to the same coset symmetry defect. More generally, for a given network of coset symmetry defects in the spacetime, redefining the sandwich $g \to k_0 g k_0^{-1}$ for all the defects in the network with a single element $k_0 \in K$ gives an another expression of the same network. However, this redundancy is not manifest in the symmetry fractionalization laws in Figure 8, 9. Let us investigate the effect of replacing $g \to^{k_0} g := k_0 g k_0^{-1}$ with $k_0 \in K$ on the symmetry fractionalization data. We will confirm that the relabeling $g \to^{k_0} g$ leads to the same symmetry fractionalization class of the coset symmetry, related by a natural isomorphism of the modular tensor category.

In Figure 8, the replacement $g \to^{k_0} g$ affects on the phase factor by

$$\bigoplus_{k \in K} \eta_{k_a}(g, h) \ ... \to \bigoplus_{k \in K} \eta_{k_a}(^{k_0}g, ^{k_0}h) \ ... \tag{5.5}$$

The above change of the phase factor is rewritten as

$$\eta_{k_a}(^{k_0}g, ^{k_0}h) = \frac{\gamma_{(k\bar{k}_0 a)}(gh)}{\gamma_{g^{-1}(k\bar{k}_0 a)}(h)\gamma_{(k\bar{k}_0 a)}(g)}\eta_{(k\bar{k}_0 a)}(g, h), \tag{5.6}$$

with

$$\gamma_x(g) = \frac{\eta_{k_0 x}(g, k_0^{-1})}{\eta_{k_0 x}(k_0^{-1}, k_0 g k_0^{-1})}. \tag{5.7}$$

This transformation by $\gamma$ is a natural isomorphism, which corresponds to redefining the $G$ action on the theory $\mathcal{C}$ by the unitary $\hat{\Gamma}_g \ket{a, b; c} = [\gamma_a(g)\gamma_b(g)/\gamma_c(g)] \ket{a, b; c}$ acting on fusion vertices of the anyons. The natural isomorphism induces the equivalence of the $G$ symmetry action on the modular tensor category $\mathcal{C}$. By relabeling $k \to k\overline{k}_0$ in the expression (5.5), the phase factor after the replacement $g \to^{k_0} g$ is given by

$$\bigoplus_{k \in K} \breve{\eta}_{k_a}(g, h) \ ... \tag{5.8}$$

where we defined the new symbol $\breve{\eta}$ related by the natural isomorphism to the original one

$$\breve{\eta}_x(g, h) = \frac{\gamma_x(gh)}{\gamma_{g^{-1}x}(h)\gamma_x(g)} \eta_x(g, h) \tag{5.9}$$

Similarly, under the replacement $g \to^{k_0} g$ Figure 9 is transformed into

$$\bigoplus_{k, k', k'' \in K} \sum_{\nu, \sigma} M_{\mathcal{D}}^{\mu\nu} U_{k_0 g}^{\nu\sigma}(^k a, ^{k'} b, ^{k''} c) M_{\overline{\mathcal{D}}}^{\sigma\rho} \ ... \tag{5.10}$$

The above phase factor $U$ is rewritten as

$$U_{k_0 g}^{\nu\sigma}(^k a, ^{k'} b, ^{k''} c) = \frac{\gamma_{k\overline{k}_0 a}(g)\gamma_{k'\overline{k}_0 b}(g)}{\gamma_{k''\overline{k}_0 c}(g)} [U_{k_0} U_g(^{k\overline{k}_0} a, ^{k'\overline{k}_0} b, ^{k''\overline{k}_0} c) U_{\overline{k}_0}]^{\nu\sigma} \tag{5.11}$$

By relabeling the group elements $\{k, k', k''\} \to \{k\overline{k}_0, k'\overline{k}_0, k''\overline{k}_0\}$, the phase factor in Figure 9 after the replacement $g \to^{k_0} g$ is expressed as

$$\bigoplus_{k, k', k'' \in K} \sum_{\nu, \sigma} \breve{M}_{\mathcal{D}}^{\mu\nu} \breve{U}_g^{\nu\sigma}(^k a, ^{k'} b, ^{k''} c) \breve{M}_{\overline{\mathcal{D}}}^{\sigma\rho} \ ... \tag{5.12}$$

where we defined the new tunneling matrix as $\breve{M}_{\mathcal{D}} = MU_{k_0}$, $\breve{M}_{\overline{\mathcal{D}}} = U_{\overline{k}_0} M_{\overline{\mathcal{D}}}$. $\breve{M}_{\mathcal{D}}, \breve{M}_{\overline{\mathcal{D}}}$ correspond to the tunneling matrices for $\mathcal{D} \times U_{k_0}, U_{\overline{k}_0} \times \overline{\mathcal{D}}$, which is the same as $\mathcal{D}, \overline{\mathcal{D}}$ due to the fusion rule of defects (3.10). It is hence expected that $\breve{M}_{\mathcal{D}} = M_{\mathcal{D}}$, $\breve{M}_{\overline{\mathcal{D}}} = M_{\overline{\mathcal{D}}}$, but detailed analysis to verify this property is left for future work. Also, the new symbol $\breve{U}$ is related to the original one by the natural isomorphism

$$\breve{U}_g^{\nu\sigma}(a, b, c) = \frac{\gamma_a(g)\gamma_b(g)}{\gamma_c(g)} U_g^{\nu\sigma}(a, b, c) \tag{5.13}$$

Therefore, the replacement $g \to^{k_0} g$ leads to the same symmetry fractionalization class related by the natural isomorphism of the data $\{U, \eta\}$.

**Additional corners at junction of sandwich** We remark that the junctions of finite-size sandwich configurations can contain three extra corners where the exteriors of the sandwich can change, i.e. additional insertion with boundary-changing topological defects. We will not consider such junctions in our discussion, and we will take the limit where the three corners coincide into the same junction in the center. For coset symmetries, we do not need to consider such junctions because there is a canonical choice of domain wall. This domain wall is simply $\mathcal{D}_{\mathrm{Rep}(K)}$, where $K$ is the minimal nonnormal subgroup (see the discussion in section 3.2).

## 5.2 Non-invertible symmetry from invertible symmetry

A general, gapped domain wall in (2+1)D topological orders can be expressed as a sandwich, consisting of an invertible domain wall in the middle region of a possibly different topological order sandwiched by two gapped non-invertible interfaces connecting the middle region to the original theory [62, 63]. The gapped domain walls between two (2+1)D TQFTs $\mathcal{C}_1, \mathcal{C}_2$ is generally described as

$$D = D_{\mathcal{A}_1} \times U \times \overline{D}_{\mathcal{A}_2}, \tag{5.14}$$

where $D_{\mathcal{A}_1}$ denotes the gapped interface that condenses the anyons specified by the condensable algebra $\mathcal{A}_1$ of a modular tensor category $\mathcal{C}_1$, and the theory obtained by anyon condensation is given by a modular tensor category $\tilde{\mathcal{C}}$. $U$ is an invertible symmetry defect of $\tilde{\mathcal{C}}$, and $\overline{D}_{\mathcal{A}_2}$ is (orientation reversal of) the defect that condenses the algebra $\mathcal{A}_2$ of $\mathcal{C}_2$ to obtain $\tilde{\mathcal{C}}$. When $\mathcal{C}_1 = \mathcal{C}_2 = \mathcal{C}$, the above defect $D$ gives a general expression for non-invertible symmetry of the (2+1)D TQFT $\mathcal{C}$.

Several comments are in order:

- The description of non-invertible coset symmetry $\tilde{U}_{[g]}$ of (2+1)D TQFT discussed in section 5.1 fits into the form (5.14), where $\mathcal{C}_1 = \mathcal{C}_2 = \mathcal{C}/K$, $\tilde{\mathcal{C}} = \mathcal{C}$, $\mathcal{A}_1 = \mathcal{A}_2 = \mathrm{Rep}(K)$. $U$ is then taken to be an invertible $G$ symmetry defect of $\mathcal{C}$.

- The expression (5.14) directly implies that the non-invertible symmetry in (2+1)D bosonic topological order exists if and only if the TQFT contains condensable bosons. This fact has been pointed out in [61].

- Let us comment on the realization of the operator $D$ in microscopic lattice models. In the lattice system, the condensed theory $\tilde{\mathcal{C}}$ is typically defined on a specific subspace $\tilde{\mathcal{H}}$ of the whole Hilbert space $\mathcal{H}$. The operator $U$ is a unitary acting within the subspace $\tilde{\mathcal{H}}$. The form of the non-invertible operator $D = D_{\mathcal{A}_1} \times U \times \overline{D}_{\mathcal{A}_2}$ is reminiscent of the singular value decomposition (SVD) of the operator $D$, where the rank of the operator $D$ is the dimension of the subspace $\tilde{\mathcal{H}}$. [8] This analogy is precise for the cosine symmetry

---

[8]To be more precise, the SVD expresses the operator $D$ in the form $D = D_1' V \overline{D}_2'$ where $V$ is a full rank diagonal matrix acting within the Hilbert space $\tilde{\mathcal{H}}$. The matrices $D_1', \overline{D}_2'$ satisfy $D_1'^\dagger D_1' = \overline{D}_2' \overline{D}_2'^\dagger = \mathrm{id}_{\tilde{\mathcal{H}}}$.

of $S_3$ gauge theory discussed in section 4. In this case, the subspace $\tilde{\mathcal{H}}$ is specified by $Z_e^b = 1$ for all edges as well as $V_{\mathbb{Z}_2}^C = 1$, which is the Hilbert space of $\mathbb{Z}_3$ toric code even under charge conjugation action $V_{\mathbb{Z}_2}^C$. We have $D_{\mathcal{A}_1} = \Pi^b \mathcal{D}^b$, $\overline{D}_{\mathcal{A}_2} = \mathcal{D}^b \Pi^b$, $U = U_\theta$. They satisfy $D_{\mathcal{A}_1}^\dagger D_{\mathcal{A}_1} = \overline{D}_{\mathcal{A}_2} \overline{D}_{\mathcal{A}_2}^\dagger = \mathrm{id}$ (up to a real positive value) within the subspace $\tilde{\mathcal{H}}$, ensuring that the cosine symmetry fits into the form of SVD.

In general, for a given non-invertible symmetry operator $D$ commuting with the Hamiltonian $DH = HD$, let us take an SVD $D = D_{\mathcal{A}_1} \times U \times \overline{D}_{\mathcal{A}_2}$. If there exists a Hamiltonian $H'$ in the subspace $\mathcal{H}'$ satisfying $HD_{\mathcal{A}_1} = D_{\mathcal{A}_1} H'$, $\overline{D}_{\mathcal{A}_2} H = H' \overline{D}_{\mathcal{A}_2}$, then we have $UH' = H'U$ and the non-invertible symmetry operator gives a sandwich expression using an invertible symmetry $U$ and topological interface operators $D_{\mathcal{A}_1}, \overline{D}_{\mathcal{A}_2}$. We conjecture that the non-invertible symmetry operator $D$ in (2+1)D topological order generally admits SVD into the composition of topological operators (5.14).

## 5.3   Bulk TQFT for finite coset non-invertible symmetry

Let us show that theories with coset symmetry $G/K$ can live on the boundary of $G$ gauge theory, where we take $G$ to be a finite group.

Let us first consider the case $K = 1$. Any quantum system with $G$ symmetry can live on the boundary of an $G$ SPT phase in the bulk. The $G$ SPT phase has a topological interface with $G$ gauge theory given by imposing the Dirichlet boundary condition of the $G$ gauge field, i.e. condensing the electric Wilson lines. By shrinking the $G$-SPT region, we find the theory can live on the boundary of a $G$ gauge theory with Dirichlet boundary condition of the $G$ gauge field. When the $G$ symmetry is anomalous, the bulk $G$ gauge theory has nontrivial topological action $\omega$ describing the anomaly. In $(D+1)$-dimensional bulk, we can consider $\omega \in H^{D+1}(BG, U(1))$. For instance, on the $e$-condensed boundary of $\mathbb{Z}_2$ gauge theory there is $\mathbb{Z}_2$ global symmetry.[9]

For general subgroup $K$, we can consider imposing a mixed boundary condition of the $G$ gauge field, where the $G$ gauge group in the bulk is broken to a subgroup $K$ on the boundary. If the bulk $G$ gauge theory has topological action $\omega$, the subgroup should satisfy

$$\omega|_K = -d\alpha \ , \tag{5.15}$$

and the boundary $K$ gauge theory has a topological action $\alpha$. There are different topological actions on the boundary of $D$ spacetime dimensions, related by shifting $\alpha \to \alpha + \nu$ with $d\nu = 0$ classified by $\nu \in H^D(BK, U(1))$. We note that the $\mathrm{Rep}(K)$ symmetry comes from the Wilson lines in the bulk $G$ gauge theory parallel to the boundary, while the magnetic operators of the $G$ gauge theory generate 0-form symmetry on the boundary.[10] See e.g. [59]

---

[9]Such boundary condition is also recently discussed in [64].

[10]The description of symmetry using bulk TQFT is also discussed in e.g. [65–67].

for a lattice construction for such bulk-boundary description. We remark that the condition (5.15) on the subgroup $K$ is also appears in e.g. [68].

### 5.3.1 Anomalies of finite coset non-invertible symmetry

Let us use the bulk TQFT to investigate whether the coset symmetry can be realized by an invertible phase, following the method of [34, 35]. In other words, we will study whether or not the 0-form coset symmetry in (2+1)D is anomalous. If the 0-form symmetry were an invertible symmetry $G$, then we would be determining whether or not the symmetry enriched topological order has a nontrivial $H^4(G, U(1))$ class. Since the coset symmetry is non-invertible, its anomaly is not labeled simply by SPTs in (3+1)D.

For coset symmetry $G/K$, we will examine the situation when the bulk TQFT is the untwisted $G$ gauge theory without any bulk topological action. We can put the bulk TQFT on an interval, on one end we impose the boundary condition corresponding to a (2+1)D theory with $G$ gauge group broken down to its non-normal subgroup $K$. This means that the $G$ Wilson lines that can end at the boundary are those whose decomposition under the subgroup $K$ contain the trivial $K$ representation, while the other Wilson lines remain nontrivial on the boundary (i.e. they are not condensed at the boundary). In addition, the magnetic operators in the $G$ gauge theory that carry $K$-holonomy can end on the boundary, since the boundary has nontrivial $K$ gauge field; the other magnetic operators cannot end on the boundary.

We need to determine whether or not there exists a boundary condition to put on the other side such that no operators can stretch between the two boundaries. Specifically, we need to determine whether or not there exists a boundary condition describing a theory with $G$ broken down to $K'$ such that

- $K' \cap K = 1$: This guarantees that there are no nontrivial magnetic operators stretching between the two boundaries.

- There are no nontrivial representations of $G$ such that its decomposition under the subgroups $K, K'$ simultaneously contains the trivial representations of $K, K'$. In other words, there are no nontrivial Wilson lines stretching between the two boundaries.

If any of the above conditions is not met, then is no (2+1)D invertible phase with coset symmetry $G/K$. The discussion can be generalized to higher spacetime dimensions.

For instance, if $G = S_3 = \mathbb{Z}_3 \rtimes \mathbb{Z}_2$ and $K = \mathbb{Z}_2$, there are three irreducible representations $1, \text{sign}, \pi$ where $\pi$ is a two-dimensional representations. The subgroup $K'$ that satisfies $K \cap K' = 1$ is $K' = \mathbb{Z}_3$. Only the trivial representation of $S_3$ simultaneously reduces to sums containing the trivial representation under both subgroups $K, K'$. Therefore, the second condition above is not met. As a result, we can conclude that the coset symmetry $S_3/\mathbb{Z}_2$ cannot be realized in a trivially gapped theory. In our example, it is realized in $S_3$ gauge

theory, which is not trivially gapped. The same discussion applies to $G = \mathbb{Z}_N \rtimes \mathbb{Z}_2$ and $K = \mathbb{Z}_2$ for $N \geq 3$.

# 6 Application: Non-Invertible Symmetry in Spin Liquids

In a microscopic model with $G$ global symmetry such that on the low energy states a subgroup $K$ does not act, i.e. there is Gauss law constraint for $K$ on the low energy subspace, the low energy effective theory can be described by $K$ gauge theory, and the symmetry is $G/K$ that acts projectively as $G$ symmetry on the electric excitations. $G$ is called the projective symmetry group (PSG), $K$ is called the invariant gauge group (IGG), and $G/K$ is the global symmetry (called the symmetry group (SG) when $K$ is a normal subgroup), which is the quotient of PSG by IGG [44].

When $K$ is a normal subgroup, $G/K$ is an ordinary group-like invertible symmetry, and the above construction is discussed extensively in the literature. This construction is particularly useful for describing quantum spin liquids with $K$ gauge theory. For example, [44] discusses $\mathbb{Z}_2$ spin liquids described by taking $K = \mathbb{Z}_2$, which is the center of the $G = SU(2)$ projective symmetry group. Here, we generalize the construction to the case when $K$ is not a normal subgroup. Then we obtain a quantum spin liquid enriched with a non-invertible global symmetry $G/K$. The symmetry is an example of the coset non-invertible symmetry discussed in section 3 and section 4.

Suppose we start with a trivially gapped system of fields with $G$ projective symmetry group and gauge a non-normal subgroup $K$. As discussed in section 3.2, the remaining $G/K$ coset global symmetry obeys the fusion rule

$$g_1 K \times g_2 K = \sum_{k \in K} \left( g_1 k g_2 k^{-1} \right) K \ , \tag{6.1}$$

where $g_i K \in G/K$ are elements of the coset. Thus the fusion rule of the coset does not obey a group multiplication law. We note that even when $g_2 = g_1^{-1}$, the fusion does produce the coset element $K$, since $g_1 k g_1^{-1}$ in general is not in the non-normal subgroup $K$.

## 6.1 "Projective" symmetry action on Wilson lines

Since we can describe the system starting with a system with $G$ symmetry, the fields that transform under the $K$ gauge group also transforms under the $G$ symmetry. Thus the gauge non-invariant fields have $G$ symmetry, while the gauge-invariant operators have $G/K$ symmetry. To see this using the formalism of section 5, we note that if $K$ is a non-normal subgroup, the action of the $G$ symmetry can relate a nontrivial $K$ Wilson line with the vacuum line.

For instance, consider a system with $G = S_3 = \mathbb{Z}_3 \rtimes \mathbb{Z}_2$ symmetry, and we gauge the $\mathbb{Z}_2$ subgroup that acts on $\mathbb{Z}_3$ by charge conjugation. On the gauge non-invariant fields, the symmetry is $S_3$, while the gauge-invariant operators see $S_3/\mathbb{Z}_2$ symmetry. To see how the symmetry acts on the Wilson lines of $\mathbb{Z}_2$ gauge theory, let us decompose the $S_3$ representations in terms of $\mathbb{Z}_2$ representations. There are two one-dimensional irreducible representations $1, \mathrm{sgn}$ of $S_3$ and one two-dimensional irreducible representation $\pi$ of $S_3$, and they decompose into $\mathbb{Z}_2$ irreducible representations $1_0, 1_1$ (the subscript is the $\mathbb{Z}_2$ charge $0, 1 \bmod 2$) as follows:

- $1 \to 1_0$, $\mathrm{sgn} \to 1_1$.

- $\pi \to 1_0 + 1_1$.

The decomposition of $\pi$ indicates that the coset symmetry can change the type of the $\mathbb{Z}_2$ Wilson line. In particular, it maps it to the vacuum line. Note that this cannot be done with any invertible symmetry, which must preserve braiding properties. This permutation of the Wilson line to the vacuum line is consistent with the "sandwich" construction for the coset symmetry given by sandwiching the $S_3$ generator by interface that condenses the $\mathbb{Z}_2$ electric charge: the Wilson line can end on the coset symmetry generator.

## 6.2 Deconfined critical points with non-invertible symmetry

Let us consider two massless scalars that together transform as the two-dimensional representation of $S_3$. Then we gauge the $\mathbb{Z}_2$ subgroup symmetry that flips the sign of one of the scalars. The theory is a $\mathbb{Z}_2$ gauge theory with coset symmetry $S_3/\mathbb{Z}_2$. The theory has a $\mathbb{Z}_2$ 1-form symmetry generated by the Wilson line. Due to the fractionalization, there is a mixed anomaly between the 1-form symmetry and the coset symmetry. To see this, we note if we condense the $\mathbb{Z}_2$ electric charge, i.e. the Wilson line becomes trivial, this implies that the coset symmetry is extended to be the $S_3$ symmetry. This implies that the theory must have deconfined excitations that carry the anomaly. In particular, there are no interactions that can drive the $\mathbb{Z}_2$ gauge field to confined phase at the critical point.

We note that before gauging the $\mathbb{Z}_2$ symmetry, the theory is a critical theory of massless scalars without deconfined excitations, and gauging the discrete symmetry modifies the spectrum by projecting out the $\mathbb{Z}_2$ odd local operators while adding a deconfined Wilson line.

# 7 Discussion and Outlook

In this work we investigate the fractionalization of coset non-invertible symmetry using field theories and lattice models. We show that the non-invertible defects can be obtained using a sandwich construction: we can build them out of invertible defects together with condensation defects. We use operators obtained in this way to explicitly derive fractionalization data on the lattice for certain examples of non-invertible symmetry fractionalization.

There are several future directions. The framework of the consistency rules described in section 5 allows us to explore the classification of new quantum spin liquids enriched by non-invertible coset symmetry. There is much to explore in the landscape of solutions to those conditions. More generally, it would be interesting to explore constraints on the fractionalization of other non-invertible symmetries beyond the coset construction. Our discussion focuses on theories in (2+1)D, but the defect sandwich construction can be generalized straightforwardly to higher spacetime dimensions. Defects in higher dimensions would have additional higher codimension structures. It would also be interesting to explore new deconfined quantum critical points with fractionalized excitations for non-invertible symmetries. Another context where the coset symmetry can arise is in the Higgs phase of $G$ gauge theory where the gauge group is broken to a non-normal subgroup $K$, which we will explore in more detail in an upcoming work.

# Acknowledgement

We thank Shu-Heng Shao for discussions and comments on a draft. P.-S.H. was supported by Simons Collaboration of Global Categorical Symmetry, Department of Mathematics King's College London. R.K. is supported by JQI postdoctoral fellowship at the University of Maryland, and by National Science Foundation QLCI grant OMA-2120757. C.Z. is supported by the Harvard Society of Fellows and the Simons Collaboration on Ultra Quantum Matter. P.-S. H. and C.Z. are also supported in part by grant NSF PHY-2309135 to the Kavli Institute for Theoretical Physics (KITP). P.-S.H. and C.Z. thank Kavli Institute for Theoretical Physics for hosting the program "Correlated Gapless Quantum Matter" in 2024 and Perimeter Institute for hosting the conference "Physics of Quantum Information" in 2024, during which part of the work is completed.

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
