# Peer review of "Fractionalization of Coset Non-Invertible Symmetry and Exotic Hall Conductance"

_SciPost Physics_

## Round 2 · Referee Report · Anonymous (Referee 1) · 2024-8-13

Report
Requested changes
-
Eq. (3.1): This is correct when the stabilizer of $g$ in $K$ is trivial. It would be good either modify the equation so that it is always correct, or be explicit about that the trivial stabilizer is assumed.
-
Page 12: The non-invertible surface operator in $U(1)_8$ was found in https://arxiv.org/abs/1012.0911 .
-
Page 31: When a subgroup $K$ acts trivially on a low-energy effective theory, the authors considered $K$ to be gauged. While this notion seems standard in the context of spin liquid, in a general context $K$ is treated as a mere decoupled factor and not gauged. Although this is supposed to be explained in the cited reference [44], it would be nice to include a concise explanation to be self-contained for broader audience.
-
In section 6, the concrete example of application is for $S_3$ case. I wonder how the $O(2)$ exotic FQH example can occur in a microscopical model (or more desirably in an experiment). It would be nice to highlight it in this section if authors know the answer.
Recommendation
Ask for minor revision

---

## Round 2 · Referee Report · Anonymous (Referee 2) · 2024-8-16

Report
I recommend the draft for publication and have the following minor comments and questions below:
Requested changes
-
Around equation (3.2), the authors discuss the general case of gauging a discrete subgroup $K$ of a $G \rtimes_\rho K$ symmetry. However, the example in equation (3.5) is not of this type. It is worthwhile either expanding the general case or explaining why the subgroup $K$ is not normal in this case. In addition, it is worthwhile to describe why gauging a non-normal subgroup leads to a non-invertible symmetry or add appropriate references.
-
Related to the previous point, I recommend citing the references [arXiv:1704.02330] and [arXiv:1712.09542], which discuss gauging discrete subgroups and non-invertible symmetries in 1+1d.
-
Does Figure 1 apply to discrete symmetries?
Recommendation
Publish (meets expectations and criteria for this Journal)

---

## Editorial Decision

unknown